# Causal Graph Learning via Distributional Invariance of Cause-Effect Relationship

**Nang Hung Nguyen**  *nanghung-nguyen@g.ecc.u-tokyo.ac.jp*
*The University of Tokyo, Japan*

**Phi Le Nguyen**  *lenp@soict.hust.edu.vn*
*Institute for AI Innovation and Societal Impact (AI4LIFE), Vietnam*
*Hanoi University of Science and Technology, Vietnam*

**Thao Nguyen Truong**  *nguyen.truong@aist.go.jp*
*National Institute of Advanced Industrial Science and Technology (AIST), Japan*

**Trong Nghia Hoang**  *trongnghia.hoang@wsu.edu*
*School of Electrical Engineering and Computer Science,*
*Voiland College of Engineering and Architecture,*
*Washington State University, Pullman, Washington, US*

**Masashi Sugiyama**  *sugi@k.u-tokyo.ac.jp*
*RIKEN, Japan*
*The University of Tokyo, Japan*

**Reviewed on OpenReview:** *https://openreview.net/forum?id=Ey38Q882Xe*

## Abstract

This paper introduces a new framework for recovering causal graphs from observational data, leveraging the observation that the distribution of an effect, conditioned on its causes, remains invariant to changes in the prior distribution of those causes. This insight enables a direct test for potential causal relationships by checking the variance of their corresponding effect-cause conditional distributions across multiple downsampled subsets of the data. These subsets are selected to reflect different prior cause distributions, while preserving the effect-cause conditional relationships. Using this invariance test and exploiting an (empirical[1]) sparsity of most causal graphs, we develop an algorithm that efficiently uncovers causal relationships with quadratic complexity in the number of observational variables, reducing the processing time by up to $25\times$ compared to state-of-the-art methods. Our empirical experiments on a varied benchmark of large-scale datasets show superior or equivalent performance compared to existing works, while achieving enhanced scalability.

## 1 Introduction

A core challenge in causal learning is finding a directed acyclic graph (DAG) that captures the cause-effect relationships between variables in an observational dataset (Zanga et al., 2022). A direct approach to uncover these causal links is through intervention which conducts more experiments to confirm whether changes in one set of variables will consequently affect the outcome distribution of another variable (Peters et al., 2016; Guo et al., 2024). However, such interventional experiments can be prohibitively expensive. Furthermore, recovering the entire causal graph requires running intervention on an exponential number of candidate

---

[1] Empirical sparsity refers to the observation that in most benchmark datasets used for causal discovery evaluation, the augmented bidirectional graph (see Theorem 7) indicating whether two nodes are in each other's Markov blanket is $p$-degenerate (as defined in Definition 4) with a small $p$.

cause-effect relationships among subsets of variables, thus rendering this approach computationally costly and impractical (Pearl, 2009).

To sidestep such expensive interventions, numerous approximation approaches have been developed to instead find an equivalence class of DAGs (Pearl et al., 2016; Peters et al., 2017), all of which are compatible with a set of statistical evidence or constraints derived from the observational data[2]. Most of these approaches are formulated as graph searches that proceed either by finding statistical evidence to eliminate incompatible graph candidates (Spirtes & Glymour, 1991; Spirtes, 2001) or by optimizing for a heuristic score defined on graphs (Hauser & Bühlmann, 2012; Rolland et al., 2022; Montagna et al., 2023). Such approaches however are either (a) not scalable due to the expensive computation cost of running statistical tests on an exponentially large number of cause-effect relationship candidates (i.e., subsets of variables); or (b) less accurate due to the heuristic nature of the score function defined over the graph space.

For examples, Spirtes & Glymour (1991) and Spirtes (2001) required performing local conditional independence tests between effect variables and their corresponding sets of causes across all candidate causal relationships, incurring a process that scales exponentially with the number of features/variables. In contrast, Hauser & Bühlmann (2012), Rolland et al. (2022), and Montagna et al. (2023) proposed direct optimization of a global heuristic score on graphs, avoiding the need to solve such constraint satisfaction problems involving an exponentially large set of local constraints. There are also other approaches in this direction which further impose simplified assumptions on the functional structure of the causal relationship (e.g., linear relationship between effects and causes perturbed with Gaussian noises). Based on those assumptions, the problem of causal learning can be formulated as a continuous optimization task and can be solved by more effective solution techniques (Kalainathan et al., 2022; Lachapelle et al., 2019; Ng et al., 2022; 2019). However, these approaches often perform less robustly when the heuristic scores or modeling assumptions do not fit well with the nature of the observational data (see Section 5).

To mitigate the aforementioned limitations, we propose a new solution perspective based on a novel causal test. Such a method avoids imposing additional assumptions on the effect-cause data generation process while also achieving better performance with affordable computation costs. Our approach leverages the invariance of the effect-cause conditional distribution $P(\text{effect} \mid \text{cause})$ to changes in the prior cause distribution $P(\text{cause})$. This inspires a principled test for causal relationships via estimating the variance of $P(X \mid \boldsymbol{Z})$ across synthetic data augmentations that reflect different cause distributions $P(\boldsymbol{Z})$. That is, $(\boldsymbol{Z} \to X)$ potentially represents a causal relationship if the data-induced $P(X \mid \boldsymbol{Z})$ does not change much[3] across different choices of $P(\boldsymbol{Z})$. This test can then be integrated with a systematic search that identifies candidates for the causal parents of all variables with quadratic complexity, achieving improved scalability and performance over previous methods. Accordingly, our proposal method relies on the contrapositive statement: if invariance does not occur, causality does not exist. Furthermore, with mild assumptions, we can recover $\text{Pa}[X]$ among the set of plausible candidates by Theorem 1. Our technical contributions that substantiate the above include:

**1.** An invariance test that reliably determines $\boldsymbol{Z} = \text{Pa}[X]$ for each effect variable $X$ and a candidate set of causes $\boldsymbol{Z}$ via checking the variance of $P(X \mid \boldsymbol{Z})$ against changes in $P(\boldsymbol{Z})$: In particular, the invariance test identifies and constructs the most informative data augmentations to reliably approximate the variance of $P(X \mid \boldsymbol{Z})$ across potential changes to $P(\boldsymbol{Z})$. This is achieved via a downsampling scheme of the observational data that modifies $P(\text{cause})$ without changing the conditional $P(\text{effect} \mid \text{cause})$ for true $(\text{cause}, \text{effect})$ tuples (Section 4.2).

**2.** A practical parent-finding algorithm that (i) adopts a previous approach (Edera et al., 2014) to find the Markov blankets of all variables using observational data; and (ii) uses this information to construct an augmented bidirectional graph for each variable whose maximal cliques correspond to its most plausible candidate parent sets: Due to the sparsity of such augmented graphs, the number of maximal cliques (correspondingly, parent candidates – Section 4.3) is quadratic in the number of variables and can be enumerated by an effective depth-first search (DFS) algorithm. The developed invariance test (Section 4.2)

---

[2]Finding the true causal graph is not possible without running intervention since different graphs can entail the same set of statistical constraints induced by observational data.

[3]The variance of the true effect-cause conditional distribution $P(\text{effect} \mid \text{cause})$ with respect to changes in $P(\text{cause})$ is theoretically zero but in practice, $P(\text{effect} \mid \text{cause})$ has to be estimated using observational data which causes (small) additional variance due to (slight) variations in its estimation across augmented datasets.

can then be used to find the true parent set among the plausible candidates for each variable, thus recovering the true causal graph.

**3.** An extensive empirical evaluation of our proposed causal discovery framework on a variety of diverse benchmark datasets including both synthetic and large-scale real-world datasets: The reported results consistently show that our framework performs better than or comparable to prior work in terms of causal discovery performance while achieving generally better scalability, with an average of up to 73.3% reduction in spurious rate and (up to) $25\times$ reduction in processing time (Section 5).

## 2    Related Works

Existing causal discovery methods that aim to recover causal graphs from observational data without using interventional data can be categorized in three main groups (Glymour et al., 2019):

First, constraint-based methods aim to recover an equivalence class of causal graphs via deriving statistical evidence from the observational data to eliminate incompatible candidates as much as possible. Their main goal is to minimize the chance of mistaking correlation for causation and hence, maximizing the reliability of the output graph. For example, Peter-Clark (PC) (Spirtes & Glymour, 1991) and Fast Causal Inference (FCI) (Spirtes, 2001; Spirtes et al., 2013) run all possible conditional independence tests between all potential (effect, cause) tuples to find the most reliable relationship candidates that pass all tests. In practice, while such approaches often produce reliable results, they do not scale well to high-dimensional datasets since the number of (effect, cause) candidate tuples often grows exponentially in the number of variables/features (Spirtes et al., 2001).

Alternatively, score-based methods instead use heuristic scores defined on graphs to reformulate causal learning as an optimization task which associate true causal graphs with those that maximize the score (Heckerman et al., 1995; Chickering, 2002; Teyssier & Koller, 2012; Solus et al., 2021). Thus, unlike constraint-based methods which cast causal learning as a constraint satisfaction task that involves an exponentially large set of local constraints, scored-based methods recast it as a global optimization task, which often admits more scalable solutions. As a result, existing score-based methods (Hauser & Bühlmann, 2012; Rolland et al., 2022; Montagna et al., 2023) often scale better to larger datasets. However, the heuristic design of the score function imposes implicit assumptions on the causal structure of data which are often violated in practice. Consequently, the reliability (i.e., not mistaking correlation for causation) of methods in this group is relatively lower than those of constraint-based methods (Section 5.1, Figures 2 and 3).

To reconcile the above conflicting goals of reliability and scalability, model-based methods adopt additional assumptions on the effect-cause generation model of the observed data (e.g., linear relationship perturbed with Gaussian noise (Zheng et al., 2018)). This often allows a provable reformulation of the true causal structure as an optimal solution to a continuous optimization task that can be solved efficiently with numerous modern and scalable machine learning algorithms (Scanagatta et al., 2015; Zheng et al., 2018; 2020). However, their performance is often not stable in scenarios where such assumptions on the data generation process do not hold, e.g., non-linear effect-cause relationships perturbed with non-Gaussian noises.

**Existing Limitations.** Overall, existing methods are either limited by expensive processing costs (as seen in constraint-based approaches), unreliable performance due to the use of a heuristic score function and greedy nature of the optimization algorithms (as with score-based methods), or inapplicable to scenarios involving unknown data-generation models (as is the case with model-based approaches). To mitigate these limitations, we investigate an alternative approach to causal learning by exploiting the invariance of the effect-cause conditional distribution across different data augmentations that induce changes to the prior distribution of causes. This inspires a highly scalable and effective causal discovery procedure, as detailed in Section 4.

## 3    Problem Formulation and Background

Let $D$ denote a dataset comprising $n$ observations $\boldsymbol{X}^{(1)}, \ldots, \boldsymbol{X}^{(n)}$ of a set $\boldsymbol{X} = (X_1, X_2, \ldots, X_d)$ of $d$ random variables. Supposing these observations are drawn independently from an unknown distribution

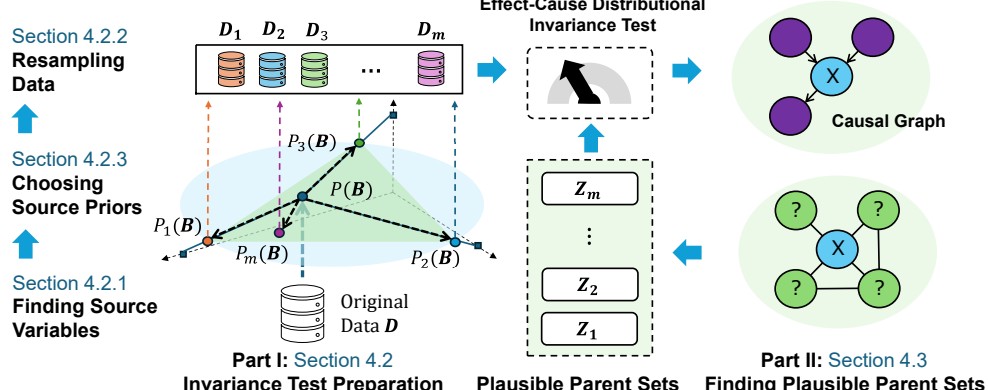

Figure 1: Overall workflow of our proposed **GLIDE** framework which comprises two main steps: (a) an algorithmic configuration — the key effect-cause distributional invariance test that helps test potential parent-child relationships (Section 4.2); and (b) a graph search algorithm exploiting prior knowledge of each node's Markov blanket and an (empirically verified) sparsity of causal graphs to provably reduce the number of tests to recover the true causal graph (Section 4.3).

$P(\boldsymbol{X}) = P(X_1, X_2, \ldots, X_d)$, we want to learn from $D$ a DAG $G = (\boldsymbol{X}, \boldsymbol{E})$ over $(X_1, X_2, \ldots, X_d)$ which is a causal model of $P(\boldsymbol{X})$ following Definition 1.

**Definition 1** (Causal Model (Peters et al., 2017))**.** A direct acylic graph $G = (\boldsymbol{X}, \boldsymbol{E})$ is a causal model of $P(\boldsymbol{X})$ if every conditional independence derived from $P(\boldsymbol{X})$ can be derived from $G$, and vice versa.

A conditional independence $X \perp\!\!\!\perp_P Y \mid \boldsymbol{Z}$ derived from $P(\boldsymbol{X})$ means $P(X \mid Y, \boldsymbol{Z}) = P(X \mid \boldsymbol{Z})$ where $P(X \mid Y, \boldsymbol{Z})$ and $P(X \mid \boldsymbol{Z})$ are marginal likelihoods derived from $P(\boldsymbol{X})$. On the other hand, a conditional independence $X \perp\!\!\!\perp_G Y \mid \boldsymbol{Z}$ derived from $G$ means given $\boldsymbol{Z}$, $X$ and $Y$ are $d$-separated following the below definition of $d$-separation.

**Definition 2** ($D$-separation (Pearl, 2009))**.** Given two variables $X, Y \in \boldsymbol{X}$ and a set of variables $\boldsymbol{Z} \subseteq \boldsymbol{X} \setminus \{X, Y\}$, $X$ and $Y$ are $d$-separated given $\boldsymbol{Z}$ (i.e., $X \perp\!\!\!\perp_G Y \mid \boldsymbol{Z}$) if any path between them contains either: (a) a fork $A \leftarrow B \rightarrow C$ with $B \in \boldsymbol{Z}$, (b) a chain $A \rightarrow B \rightarrow C$ such that $B \in \boldsymbol{Z}$; and (c) a collider $A \rightarrow B \leftarrow C$ such that $B$ or any of its descendants $\mathrm{Desc}[B]$ is not in $\boldsymbol{Z}$.

Definition 1 implies $G$ is a causal model of $P(\boldsymbol{X})$ when $X \perp\!\!\!\perp_P Y \mid \boldsymbol{Z} \Leftrightarrow X \perp\!\!\!\perp_G Y \mid \boldsymbol{Z}$. Thus, supposing $G$ is a causal model of $P(\boldsymbol{X})$, $X \perp\!\!\!\perp_P (\boldsymbol{X} \setminus \mathrm{Desc}[X]) \mid \mathrm{Pa}[X]$ since $X \perp\!\!\!\perp_G (\boldsymbol{X} \setminus \mathrm{Desc}[X]) \mid \mathrm{Pa}[X]$ due to $d$-separation – see Definition 2 – where $\mathrm{Pa}[X]$ denotes the set of parents of $X$ in $G$.

**Local Markov Conditions.** The above means that each node $X$ is independent of its non-descendants $\boldsymbol{X} \setminus \mathrm{Desc}[X]$ given its parents $\mathrm{Pa}[X]$ according to $P(\boldsymbol{X})$, resulting in the following factorization: **(I)** $P(\boldsymbol{X}) = \prod_{i=1}^{d} P(X_i \mid \mathrm{Pa}[X_i])$ which implies **(II)** $X \perp\!\!\!\perp_P (\boldsymbol{X} \setminus \mathbb{M}(X)) \mid \mathbb{M}(X)$, where $\mathbb{M}(X)$ denotes the Markov blanket of a variable $X$ that consists of its immediate parents, its children, and its children's other parents. **(II)** can also be verified via checking $d$-separation on $G$.

**Core Idea.** Supposing that Markov blanket in condition **(II)** is known, an augmented bidirectional graph for each variable $X$ can be constructed such that each of its maximal cliques corresponds to a plausible candidate $\boldsymbol{Z}$ of the true causal parents $\mathrm{Pa}[X]$. We can leverage this representation to develop an effective DFS procedure that systematically enumerates through each candidate with a quadratic time complexity in the number of variables (Section 4.3). Each candidate can be tested using the developed causal test via synthetic data augmentation (Section 4.2).

For this, we adopt an existing algorithm that can recover the Markov blankets for all $X \in G$ from observational data $D$ with $O(d^2)$ complexity (Edera et al., 2014). This is guaranteed under the causal sufficient assumption (Pearl et al., 2016) that there is no unobserved common causes (i.e., confounders) that affect the causal mechanism generating the observed data.

## 4  Invariance of $P(\textbf{Effect}|\textbf{Cause})$

First and foremost, we state our premise regarding the invariance of the causal relationships, which is well aligned with principles of causality discussed in (Peters et al., 2016) and is commonly accepted as an intrinsic property in Bayesian structures:

**Observation 1** (Causal Invariance). *Let $X \in \boldsymbol{X}$ be any variable and $\boldsymbol{B} \subset \boldsymbol{X}$ be the set of source variables[4]. The causal conditional distribution $P(X \mid \mathrm{Pa}[X])$ remains invariant with respect to changes of the joint distribution $P(\boldsymbol{B})$ over the source variables $\boldsymbol{B}$.*

**Remark 1.** *A direct implication of Observation 1 is that the conditional distribution of any variable conditioned on its parents remains invariant with respect to the change in the joint distribution $P(\boldsymbol{X})$ if the change is originated from $P(\boldsymbol{B})$ since $P(\boldsymbol{X}) = P(\boldsymbol{B}) \cdot \prod_{X \notin \boldsymbol{B}} P(X \mid \mathrm{Pa}[X])$ (local Markov Conditions).*

Relying on this logic, we design our proposed framework accordingly. To commence with, our overview is presented in Section 4.1 where we devise an efficient test to identify the causal parents of any variable. However, utilizing such a method requires that we can simulate conditional distributions across different joint distributions $P(\boldsymbol{X})$. As we will show in Section 4.2, this can be done by sampling environments of distinct joint distributions of the source variables $P(\boldsymbol{B})$. For this, we design an approximation search method to locate the source variables, then we devise a sampling strategy that is both theoretically sound and efficient. Finally, we present a plausible parent selecting strategy in Section 4.3. The bird's eye view of our workflow is depicted in Figure 1.

### 4.1  An Overview

The core idea of our causal discovery framework is based on the invariance of the (marginal) effect-cause conditional distribution $P(X \mid \mathrm{Pa}[X])$ induced from $P(\boldsymbol{X})$ across different choices of the prior $P(\boldsymbol{B})$ over the source variables $\boldsymbol{B} \triangleq \{B \mid B \in \boldsymbol{X} , \mathrm{Pa}[B] = \emptyset\} \subseteq \boldsymbol{X}$ of the true causal graph $G = (\boldsymbol{X}, \boldsymbol{E})$ — see Theorem 1 below (a detailed proof is provided in Appendix A.1).

**Theorem 1** (Effect-Cause Distributional Invariance). *Let $P_1(\boldsymbol{B}), P_2(\boldsymbol{B}), \ldots, P_m(\boldsymbol{B})$ denote a set of $m$ different priors over $\boldsymbol{B}$. Let $P_i(\boldsymbol{X}) = P_i(\boldsymbol{B}) \cdot P(\boldsymbol{X} \setminus \boldsymbol{B} \mid \boldsymbol{B})$ denote the corresponding augmentation of the true data distribution $P(\boldsymbol{X})$ when we replace its marginal prior $P(\boldsymbol{B})$ with $P_i(\boldsymbol{B})$. For each variable $X \in \boldsymbol{X} \setminus \boldsymbol{B}$, its true causal parents $\mathrm{Pa}[X]$, its spouses $\mathrm{Sp}[X]$, and plausible candidate set $\mathcal{Z} = \{\boldsymbol{Z}_1, \boldsymbol{Z}_2, \ldots\}$, assuming that $\mathrm{Pa}[X] \cap \mathrm{Sp}[X] = \mathrm{Sp}[X] \cap \boldsymbol{B} = \emptyset$, we have:*

$$\exists \boldsymbol{Z}_j \in \mathcal{Z} : \mathbb{V}_{P_i(\boldsymbol{X}) \sim \mathcal{P}} \Big[ P_i \Big( X \mid \boldsymbol{Z}_j \Big) \Big] = 0 \quad \Leftrightarrow \quad \boldsymbol{Z}_j = \mathrm{Pa}[X] \tag{1}$$

*and $P_i(\boldsymbol{X})$ is drawn from $\mathcal{P} \triangleq (P_1(\boldsymbol{X}), P_2(\boldsymbol{X}), \ldots, P_m(\boldsymbol{X}))$. Here, $\mathbb{V}_{P_i(\boldsymbol{X}) \sim \mathcal{P}} [P_i(X \mid \boldsymbol{Z}_j)]$ denotes the statistical variance of the induced $P_i(X|\boldsymbol{Z}_j)$ over the random choice of the joint distribution $P_i(\boldsymbol{X})$.*

Intuitively, we argue that the parent set of a variable $X$ is one of those when conditioned on, the probability of observing $X$ becomes unchanged, hence zero statistical variance with respect to distributional changes of the source variables. With mild assumptions (see Appendix A.1), we can assure the uniqueness of such $\boldsymbol{Z}_j$ in Equation 1. This result reveals a principled test of whether a subset $\boldsymbol{Z}_j \in \mathcal{Z} \subseteq \boldsymbol{X}$ is the parent set of $X \in \boldsymbol{X}$ according to the true causal graph $G$. The intuition is that if we have access to $D_1, D_2, \ldots D_m$ such that $D_i \sim P_i(\boldsymbol{X})$, we can test whether $\boldsymbol{Z}_j = \mathrm{Pa}[X]$ via checking whether the sample variance of the empirical conditional $P_i(X \mid \boldsymbol{Z}_j)$ is distinguishably small. As we show in Theorem 5 later, the aforementioned $\{D_i\}_{i=1}^m$ can be simulated by sampling from the observational data $D$. Further details on the sampling method is discussed in Section 4.2.2. Given $D_1, D_2, \ldots D_m$ above, we perform the invariance test as formalized below:

**A. Effect-Cause Invariance Test.** Given $m$ augmented datasets $D_1, D_2, \ldots, D_m$ which are resampled from $D \sim P(\boldsymbol{X})$ such that $D_i \sim P_i(\boldsymbol{X})$, then $\boldsymbol{Z}_j \in \mathcal{Z}$ is the causal parent set of $X$, i.e., $\boldsymbol{Z}_j = \mathrm{Pa}[X]$, when

$$\frac{1}{m} \sum_{i=1}^m \Big\| P_i \Big( X \mid \boldsymbol{Z}_j \Big) - \overline{P} \Big( X \mid \boldsymbol{Z}_j \Big) \Big\|^2 \simeq 0, \quad \text{where} \quad \overline{P} \Big( X \mid \boldsymbol{Z}_j \Big) = \frac{1}{m} \sum_{i=1}^m P_i \Big( X \mid \boldsymbol{Z}_j \Big) . \tag{2}$$

---

[4]A source variable/node (a.k.a. root node, exogenous node) is a node that has no parent nodes

Empirically, we consider $\simeq 0$ equivalent to $< \varepsilon$ where $\varepsilon \sim 10^{-3}$. This test will become more accurate with more source priors. When $m$ is sufficiently large and $\{P_i(\boldsymbol{B})\}_{i=1}^m$ are sufficiently representative of the entire space of source priors, the test might not be perfect but remains highly accurate (see Section 5). In the case multiple $\boldsymbol{Z}_j \in \mathcal{Z}$ satisfy Equation 2, the one with minimum variance is chosen.

**B. High-Level Framework.** Supposing that we know how to generate $D_1, D_2, \ldots, D_m$ (see Section 4.2) for which $D_i \sim P_i(\boldsymbol{X})$ as required in Theorem 1, solving Equation 2 can be achieved via exhaustively checking all subsets $\boldsymbol{Z}$ as candidates for $\mathrm{Pa}[X]$. Repeating this for all $X$ allows us to recover the causal graph. This process is, however, impractical since its complexity is exponential in $d$.

Fortunately, this can be avoided using the local Markov condition **(II)** (see Section 3) which implies that (i) $\boldsymbol{Z} \subseteq \mathbb{M}(\boldsymbol{X})$ if $\boldsymbol{Z} = \mathrm{Pa}[X]$ and (ii) there are at most $p = O(d)$ candidates for $\mathrm{Pa}[X]$ which can be provably identified via finding maximum cliques on augmented bidirectional graphs. Under an empirically verified assumption on causal graph sparsity, this can be achieved with a customized DFS procedure with $O(d^2)$ complexity (see Theorem 7, Section 4.3). This results in an $O(d^2)$ total complexity for our causal discovery framework as detailed below.

**C. Time Complexity.** As there are $d$ observational variables, our framework needs to make $d$ calls to the parent-finding routine in part (**B**). As each routine will consider at most $p = O(d)$ plausible parent candidates, the per-call complexity is $O(md)$. This amounts to a total cost of $O(md^2)$ to recover the full causal graph. This can be enabled with (a) a practical $O(d^2)$ overhead to find the Markov blankets for all variables following the prior work of Edera et al. (2014); and (b) another $O(d^2 + m|D| + m|\boldsymbol{B}|)$ overhead to construct $D_1, D_2 \ldots, D_m$ for Equation 2 (Section 4.2.4). The overall complexity (including the overhead) is therefore $O(md^2 + m|D| + m|\boldsymbol{B}|)$.

**D. Identifiability.** Generally, our proposed method does not guarantee a unique solution given solely the observational dataset. However, assuming that the basis set found by Theorem 2 matches the set of source variables, **GLIDE** can recover the correct causal parent set for all nodes that satisfy Theorem 1's assumptions. In practice, we can only estimate the basis set (Theorem 3) and there exist variables who do not satisfy assumptions in Theorem 1. Regardless, as shown in Section 5, our performance remains superior over state-of-the-art (SOTA) baselines.

To substantiate the above high-level framework, we need to (i) choose representative variants of the source priors to improve the test reliability (Section 4.2); and (ii) find all plausible sets of candidate for $\mathrm{Pa}[X]$ for all $X \in \boldsymbol{X}$ which are guaranteed to have sizes at most $O(d)$ (Section 4.3).

## 4.2 Augmenting Source Prior

To enable practical use of the parent-finding routine in Equation 2, we need to (i) find the set $\boldsymbol{B}$ of sources, (ii) choose a representative set of source priors $P_1(\boldsymbol{B}), P_2(\boldsymbol{B}) \ldots, P_m(\boldsymbol{B})$ so that the invariance test is reliable; and (iii) re-sample $D_i$ from $D$ so that $D_i \sim P_i(\boldsymbol{X}) = P_i(\boldsymbol{B}) \cdot P(\boldsymbol{X} \setminus \boldsymbol{B} \mid \boldsymbol{B})$.

The above (iii) is essential because we do not have direct access to $P(\boldsymbol{X} \setminus \boldsymbol{B} \mid \boldsymbol{B})$. This means that we cannot compute $P_i(\boldsymbol{X})$ and induce $P_i(X \mid \boldsymbol{Z})$ to assess its variance directly. However, as $P(\boldsymbol{X} \setminus \boldsymbol{B} \mid \boldsymbol{B})$ can be simulated using the original data $D$, we can re-sample $D_i$ from $D$ so that $D_i \sim P_i(\boldsymbol{X})$, which then allows us to use $D_i$ to estimate $P_i(X \mid \boldsymbol{Z})$ and consequently, its variance.

For this, we will develop an algorithm to find the source variables $\boldsymbol{B}$ (Section 4.2.1). We will then derive a re-sampling procedure to obtain a downsampled dataset $D_i$ from $D$ such that $D_i \sim P_i(\boldsymbol{X})$ (Section 4.2.2). Last, we will detail a criterion to choose informative $P_i(\boldsymbol{B})$ (for the invariance test) based on the above re-sampling procedure and how to determine a set of representative source priors (Section 4.2.3).

### 4.2.1 Finding Source Variables

To find the set of source variables (i.e., those with no parent in the true causal graph), we introduce below the concept of a basis of a DAG.

**Definition 3.** A basis $\boldsymbol{B} \subseteq \boldsymbol{X}$ of a DAG $G = (\boldsymbol{X}, \boldsymbol{E})$ is a set of mutually $d$-separated (independent) variables (see Definition 2) such that for each $X \notin \boldsymbol{B}$, there exists $X' \in \boldsymbol{B}$ such that $X' \not\perp\!\!\!\perp X$.

As finding the true sources is not possible without intervention, we will use a basis set as a surrogate. This is because the basis has similar properties to the set of sources. First, similar to source variables, basis variables are mutually independent. Second, each source variable $X$ is either in the basis $\boldsymbol{B}$ or shares the same dependence set $\Phi(X) \equiv \Phi(X')$ — as defined in Theorem 3 — with a basis variable $X' \in \boldsymbol{B}$ (see Appendix B.2). Furthermore, Theorem 2 confirms that the maximum size of a basis set is equal to the number of sources in the true causal graph. This means that changing the prior over basis variables will have similar effect to changing priors over source variables. Hence, we can use the priors over basis variables instead of priors over sources to test the effect-cause invariance.

**Theorem 2.** *The maximum size of a basis set of a DAG is upper-bounded by the number of its sources.*

The proof of Theorem 2 is deferred to Appendix A.2. A direct corollary of Theorem 2 is that the maximum sized basis has the same cardinality as the source set of the graph. Theorem 3 provably provides a systematic method to identify such a maximum sized basis set with $O(d^2)$ computational complexity. See Appendix A.3.

**Theorem 3.** *Let $\boldsymbol{V} = \boldsymbol{X}$ and $\Phi(X) \triangleq \{Y \mid Y \in \boldsymbol{V} : Y \not\perp X\}$. A set $\boldsymbol{B}$ is constructed via iteratively (i) inserting into $\boldsymbol{B}$ variable $X \in \boldsymbol{V}$ with the lowest $|\Phi(X)|$; (ii) setting $\boldsymbol{V} \leftarrow \boldsymbol{V} \setminus \Phi(X)$; and (iii) stopping when $\boldsymbol{V} = \emptyset$. Then, $\boldsymbol{B}$ forms a maximum-sized basis.*

Note that $X \in \Phi(X)$ by default since $X \not\perp X$, $\forall X$. The intuition of Theorem 3 is that given a causal graph, the set of dependence nodes for a source does not include other sources while most non-source nodes are dependent on all their upstream (including all source nodes) and downstream nodes. Thus, the set of dependence nodes for a source node will not be larger than that of a non-source node. Computing $\Phi(X)$ requires checking $Y \not\perp X$ which can be practically achieved using existing reliable pairwise independence tests provided by the causal-learn open library (Zheng et al., 2024). The independence checking for $(Y \not\perp X)$ is done by the $O(n)$ Fisher test (Fisher et al., 1921). As there are $O(d^2)$ such pairs, the total cost to compute all $\Phi(X)$ is $O(nd^2)$. Furthermore, $\Phi(X)$ does not need recomputing after step (ii) in Theorem 3. Hence, computing $\Phi$ is a one-time overhead that is not expensive. It is worth noting that the non-uniqueness of the basis set has negligible empirical impact on the overall performance of our proposal. In fact, across multiple random seeds, data generating models, and topologies, the highest recorded variance in term of SHD is less than 6% of the mean value across a variety of experimental scenarios.

### 4.2.2 Re-sampling Observational Data

Given a target $P_i(\boldsymbol{B})$, we want to find a resampled dataset $D_i$ from the original observation data $D$ such that $D_i \sim P_i(\boldsymbol{X})$. To guarantee this, $D_i$ must be a downsampled version of $D$ via sampling with no replacement to avoid introducing duplicates and hence, false causal biases into $D_i$. Intuitively, because this operation only removes information rather than generating new samples, it cannot introduce any spurious patterns into the data. To elaborate, downsampling can be viewed as introducing an auxiliary selection variable for each observational snapshot of the underlying DAG. Spurious dependence can arise only if this selection variable conditions on a collider of the form $U \to X \leftarrow V$ where $U$ and $V$ originate from different source ancestries, in which case the corresponding trail changes from inactive to active. In our construction, however, downsampling uses $\boldsymbol{B} = \boldsymbol{b}$ as the sole selection criterion, so the selection variable depends only on basis variables. Each basis variable is either a source or a node with single-source ancestry (see Lemma 2). In the latter case, such a basis node can participate in a collider $U \to X \leftarrow V$ only when $U$ and $V$ share the same source ancestry; otherwise, $X$ would depend on multiple sources and, by Lemma 2, could not be selected into the basis. Since $U$ and $V$ are already dependent via their common source, activating this collider does not introduce any new dependence. On the other hand, among valid downsamples $D_i \sim P_i(\boldsymbol{X})$, we want to choose the one that has the minimal downsampled rate $|D|/|D_i|$ to preserve (as much as possible) observations of the effect-cause conditionals in $P(\boldsymbol{X} \setminus \boldsymbol{B} \mid \boldsymbol{B})$. Interestingly, it can be shown that given the target $P_i(\boldsymbol{B})$, the minimum downsampling rate can be computed and achieved via the following results.

**Theorem 4.** *Suppose that $D_i$ is the downsampled dataset with minimum downsampling rate that satisfies the condition $D_i \sim P_i(\boldsymbol{X})$. Then, it follows that*

$$|D|/|D_i| = \gamma_i^{-1}, \quad where \quad \gamma_i = \min_{\boldsymbol{b}} \left( P(\boldsymbol{B} = \boldsymbol{b})/P_i(\boldsymbol{B} = \boldsymbol{b}) \right) . \tag{3}$$

Given the (computable) optimal downsampling rate in Theorem 4, the corresponding downsampling procedure that achieves it can be derived via Theorem 5 below.

**Theorem 5.** *Let $\gamma_i$ be defined in Theorem 4. Suppose that $D_i$ is created via sampling without replacement $P_i(\boldsymbol{B} = \boldsymbol{b}) \cdot |D| \cdot \gamma_i$ points from $D$ where $\boldsymbol{B} = \boldsymbol{b}$. Then, $|D|/|D|_i = 1/\gamma_i$ and $D_i \sim P_i(\boldsymbol{X})$.*

The proofs of Theorems 4 and 5 are deferred to Appendices A.4 and A.5. To elaborate more on the sampling of $D_i$ from $D$, we determine the size of $D_i$ via Theorem 4. This allows us to compute (for each candidate value $\boldsymbol{b}$ of the source variables $\boldsymbol{B}$) how many samples (where $\boldsymbol{B} = \boldsymbol{b}$) we need to draw (without replacement) from $D$. Those samples (across all observed candidate values $\boldsymbol{B} = \boldsymbol{b}$) constitute $D_i$. Then, Theorem 5 guarantees that such $D_i$ is statistically equivalent to a direct sample from $P_i$. In a practical implementation, we associate $P(\boldsymbol{B} = \boldsymbol{b})$ with its empirical estimate $P(\boldsymbol{B} = \boldsymbol{b}) \approx |D[\boldsymbol{B} = \boldsymbol{b}]|/|D|$. For continuous data, we categorize the input data into fixed-size bins.

### 4.2.3 Choosing Source Priors

We can now leverage the insight of Theorem 4 to choose the most informative source priors to enhance the reliability of the invariance test in Equation 2. Intuitively, we want $D_i \neq D$ so $\gamma_i$ should not be too large. Otherwise, as $\gamma_i \to 1$, $D_i \to D$, and $P_i(\boldsymbol{B}) \to P(\boldsymbol{B})$ which cannot be used to test the effect-cause invariance. On the other hand, if $\gamma_i$ is too small, $D_i$ might drop too much information from $D$ which might obscure some effect-cause relationship.

As such, we want to choose $P_i(\boldsymbol{B})$ such that its inverse downsampling rate $\gamma_i$ (see Theorem 4) is above a certain threshold $\gamma_{\mathrm{o}}$ where $\gamma_{\mathrm{o}} \in (0, 1)$ is an adjustable parameter. Our ablation studies in Table 3 investigate the impact of $\gamma_{\mathrm{o}}$ on the overall causal discovery performance. The question is now how to sample representative $P_i(\boldsymbol{B})$ from the subspace of source priors whose (inverse) optimal downsampling rate $\gamma_i \geq \gamma_{\mathrm{o}}$. To answer this question, we establish Theorem 6 below which characterizes its convex hull (see Appendix A.6).

**Theorem 6.** *Let $r \triangleq |\mathrm{Dom}(\boldsymbol{B})|$ denote the number of (categorical) candidate values of $\boldsymbol{B}$. The subspace of $P_i(\boldsymbol{B})$ that satisfies $\gamma_i \geq \gamma_{\mathrm{o}}$ is a convex subspace $C_r(\gamma_{\mathrm{o}})$ of the $r$-dimensional simplex $\Delta_r$ over $\mathrm{Dom}(\boldsymbol{B})$ which cuts $\Delta_r$ at $r$ points $P^{(1)}(\boldsymbol{B}), P^{(2)}(\boldsymbol{B}) \dots, P^{(r)}(\boldsymbol{B})$ representing its convex hull:*

$$P^{(k)}(\boldsymbol{B}) \quad = \quad \alpha_k \cdot P(\boldsymbol{B}) + (1 - \alpha_k) \cdot \delta_k(\boldsymbol{B}) \quad where \quad \alpha_k \quad = \quad (1 - q_k \gamma_o^{-1})/(1 - q_k) , \tag{4}$$

*$q_k = P\big(\boldsymbol{B} = \boldsymbol{b}^{(k)}\big)$, and $\delta_k(\boldsymbol{B})$ is a point mass function that assigns $1$ when $\boldsymbol{B} = \boldsymbol{b}^{(k)}$ and $0$ otherwise. Here, $\boldsymbol{b}^{(k)}$ is the $k$-th candidate value in $\mathrm{Dom}(\boldsymbol{B})$.*

Each source prior $P_i(\boldsymbol{B})$ with $\gamma_i \geq \gamma_{\mathrm{o}}$ belongs to this convex set and can be represented as a linear combination of the above points:

$$P_i(\boldsymbol{B}) \quad = \quad \sum_{k=1}^{r} a_k \cdot P^{(k)}(\boldsymbol{B}), \tag{5}$$

where $\sum_{k=1}^{r} a_k = 1$ and $a_k \geq 0$. Hence, $P_i(\boldsymbol{B})$ can be sampled via drawing $\boldsymbol{a} = (a_1, a_2, \dots, a_r)$ from the simplex $\Delta_r$ and using Equation 5.

### 4.2.4 Practical Implementation

The parent-finding procedure involves three main steps: finding basis variables, sampling source priors, and re-sampling observational data. First, finding the basis variables has a time complexity of $O(d^2)$. Next, we sample $\boldsymbol{a}$ from a Dirichlet distribution to compute $P_i(\boldsymbol{B})$, with $10^4$ samples clustered using $K$-means. The $K = m$ centroids are selected as source priors, with a complexity of $O(mr) = O(mc^{|\boldsymbol{B}|})$ where $c$ is the maximum number of candidate values of a single variable. To avoid exponential computation costs in $|\boldsymbol{B}|$, we resample for each basis variable individually, reducing the complexity to $O(mc|\boldsymbol{B}|)$. Finally, generating the augmented dataset for each $P_i(\boldsymbol{B})$ incurs a cost of $O(m|D|)$ via Theorem 5. This amounts to the $O(d^2 + mc|\boldsymbol{B}| + m|D|)$ total time complexity. As for hyper-parameters, we recommend choosing $m$ as high as the incurred additional runtime is acceptable. As for $\gamma$, we find $\gamma = 0.5$ is an empirically reasonable selection. As a rule of thumb, $\gamma$ should be high enough so that estimations induced from downsampled data are adequately accurate, but also should be small enough to promote diversity between downsampled datasets.

### 4.3 Finding Plausible Parent Sets

To ensure that the parent-finding routine is scalable, we customize a DFS algorithm which leverages prior work (Edera et al., 2014) in Markov blanket identification to provably find $O(d)$ candidate parent sets for each effect variable $X$. The true causal graph can then be recovered via solving Equation 1 for each $X$ with respect to the discrete set of $O(d)$ plausible parents found above.

As previous work in Markov blanket identification incurs an $O(d^2)$ cost (Edera et al., 2014), the total cost of our parent finding phase is also $O(d^2)$. This helps avoid the brute-force search through all subsets of variables as candidates for $\mathrm{Pa}[X]$ whose complexity is otherwise exponential in $d$ (Peters et al., 2016). To achieve this, we leverage the following result which relates candidate sets of causal parents to maximal cliques on an augmented bidirectional graph.

**Theorem 7** (Plausible Parent Sets). *Let $\mathbb{M}(X)$ denote the Markov blanket of $X$ and $\mathbb{M}^*(X) \triangleq \mathbb{M}(X) \setminus \mathrm{Sp}[X]$ denote the set of non-spouse variables within $\mathbb{M}(X)$. For each variable $X$, let $G'(X) = (\mathbf{V}, \mathbf{E}')$ denote a bidirectional graph where $\mathbf{V} = \mathbb{M}^*(X)$ and $(U, V) \in \mathbf{E}$ iff $V \in \mathbb{M}(U)$ and $U \in \mathbb{M}(V)$. Then, $\mathrm{Pa}[X] \subseteq \mathbb{M}^*(X)$ belongs to at least a clique in $G'(X)$.*

Here, a clique is defined to be a set of nodes in the bidirectional graph $G'$ such that there is an edge between any two nodes. See Appendix A.7 for a detailed proof. Notice that $\mathbb{M}^*(X)$ can be derived directly from $\mathbb{M}(X)$ by using conditional independence tests. Indeed, since $\forall Y \in \mathrm{Sp}[X] : Y \perp\!\!\!\perp X \mid \mathrm{Pa}[X]$ and $\mathrm{Pa}[X] \subset \mathbb{M}(X)$, we are guaranteed to detect and remove all such $Y \in \mathbb{M}(X)$, hence $\mathbb{M}^*(X)$. Finding $\mathbb{M}^*(X)$ for any $X \in \mathbf{X}$ costs a computational complexity of $O(2^M)$ where $M \ll d$ is the maximum number of elements in a Markov blanket. Leveraging Theorem 7, we can find all plausible parent sets of $X$ as subsets in $\mathbb{M}^*(X)$, each corresponding to a clique in $G'(X)$. Restricting the set of plausible parent sets to that of maximal cliques in $G'$, we can reduce the task of finding plausible parent sets to finding maximal cliques in a bidirectional graph. For a sparse graph $G'(X)$ with a low degeneracy constant $p$ (see Definition 4), this can be achieved effectively via a customized version of the DFS-based Bron-Kerbosch (Bron & Kerbosch, 1973) algorithm (see Appendix B.3) which finds all maximal cliques with $O(dp3^{p/3})$ time complexity.

**Definition 4** (Degeneracy (Kirousis & Thilikos, 1996)). A bidirectional graph is $p$-degenerate if every subgraph has at least one node with degree less than or equal to $p$. The graph's degeneracy is the smallest $p$ for which it is $p$-degenerate.

Considering $p$ as a small constant compared to $d$, the cost of finding all plausible parent sets for each variable $X$ is effectively $O(d)$. Hence, the total cost of finding all plausible parent sets for all variables is $O(d^2)$. Furthermore, it is also established in Bron & Kerbosch (1973) that the (worst-case) number of maximum cliques is $O\big((d - p) \cdot 3^{p/3}\big)$. Our empirical studies in Appendix C.4.3 show that $p \leq 13$ across all benchmark datasets even for the largest graphs, confirming that the set of maximal cliques in $G'(X)$ is practically $O(d)$ with a constant factor smaller than $3^{13/3} < 117$.

## 5 Experiment

This section evaluates and compares the performance of our proposed method, Causal Graph Learning via Distributional Invariance of Cause-Effect Relationship (**GLIDE**), against existing state-of-the-art (SOTA) baselines in causal graph learning.

**Baselines.** Our empirical evaluations are conducted with respect to a variety of data models and a diverse suite of both classical and recent baselines (Appendix C.1 Table 1), including PC (Spirtes & Glymour, 1991), GIES (Hauser & Bühlmann, 2012), FCI (Spirtes, 2001), NOTEARS (Zheng et al., 2018), MLP-NOTEARS (Zheng et al., 2020), DAS (Montagna et al., 2023), and SCORE (Rolland et al., 2022). Each baseline is configured with its best hyper-parameters (see Appendix C.1).

**Datasets.** Our experiments are based on both synthetic and real-world datasets. For synthetic experiments (Sections 5.1 and Appendix C.2), we follow the commonly used protocols in previous work to generate observationals data based on the Erdos-Renyi (Zheng et al., 2018; Heinze-Deml et al., 2018), bipartite and scale-free (Zheng et al., 2020) classes of causal graphs. Due to limited space, we only present the empirical

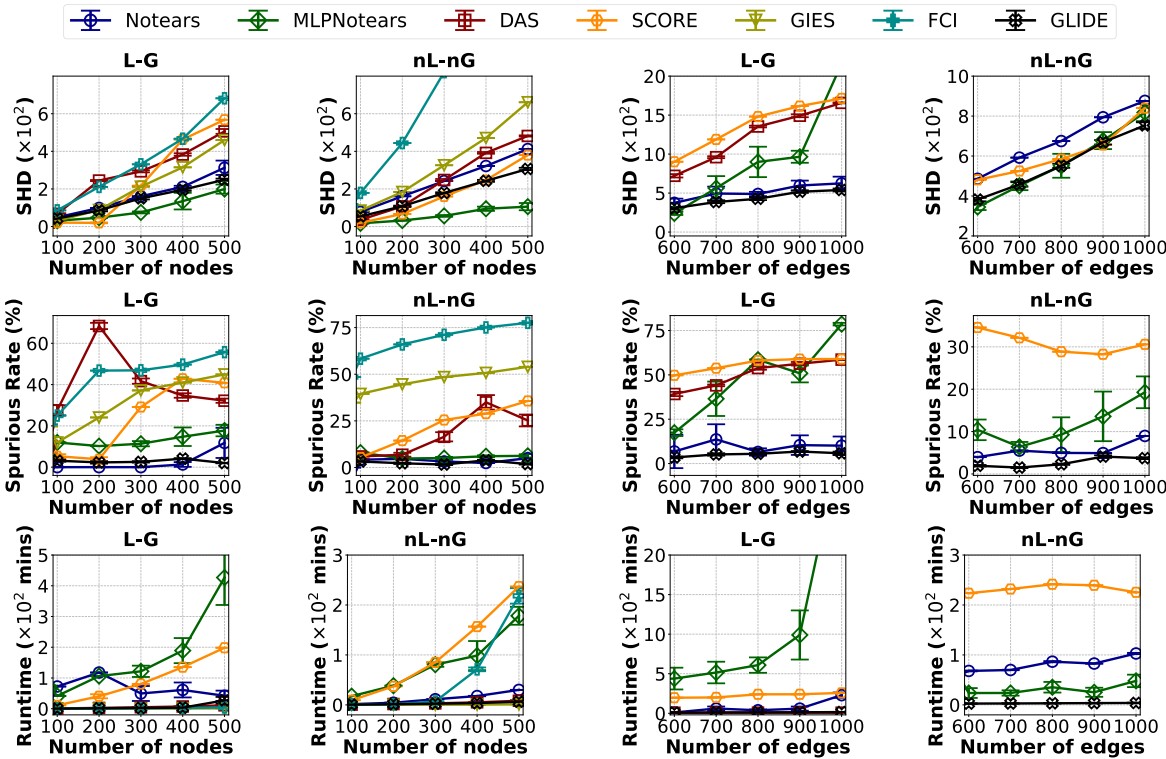

Figure 2: Baseline performance in normal setting (continuous data). Lower metrics are better.

Figure 3: Baseline performance in extreme setting (continuous data). Lower metrics are better.

results on synthetic data generated from the Erdos-Renyi class of graphs. Other experiments with the bipartite and scale-free classes of graphs are deferred to Appendix C.4. We also provide evaluation on the SACHS dataset (Section 5.2) provided by Sachs et al. (2005). This dataset has been widely used in many existing literatures on causal discovery Zheng et al. (2018; 2020). The SACHS dataset (11 variables, 17 edges) contains records of continuous measurements of expression levels of proteins and phospholipids in human immune system cells and is widely accepted by the biology community. Additionally, we also conduct experiments on real-world graphs (see Appendix C.3) using the datasets provided in the bnlearn package (Scutari, 2009), which includes 7 small to large real-world graphs. Notably, the Munin dataset features a large-scale graph comprising 1041 variables.

**Evaluation Metrics.** We use the structural Hamming distance (SHD), spurious rate (percentage), and running time (minutes) to measure both the effectiveness and scalability of our proposed method. The SHD and running time metrics are commonly used in the evaluation of existing causal discovery methods (Zheng et al., 2018; Montagna et al., 2023). The spurious rate measures the ratio of the number of false causal relationships among all causal relationships discovered by each algorithm. This helps compare the reliability of causal prediction across baselines. All performance metrics are reported with their mean and confidence interval 95% averaged over 10 independent runs.

## 5.1 Synthetic Continuous Data

We consider two experiment settings. First, in the normal setting, the graph's number of edges is the same as its number of nodes, which varies from 100 to 500. Second, in the extreme setting, we fix the number of nodes at 500 and increase the number of edges from 600 to 1000. In each setting, we further consider two data-generation models for the effect-cause relationship: (1) linear relationship perturbed with Gaussian noise (annotated as **L-G**); and (2) non-linear and non-Gaussian relationship (annotated as **nL-nG**). Such data generation mechanism is implemented using the public code in (Zheng et al., 2018; 2020). In each experiment, we first generate a random Erdos-Renyi DAGs and then simulate (synthetic) observational data from it using the above mechanisms. Our reported results and observations in each setting are detailed below.

**Normal Setting.** Figure 2 reports the performance of our method **GLIDE** in comparison to those of other baselines. In the **L-G** cases, **GLIDE**, NOTEARS, and MLP-NOTEARS outperform the rest of the baselines significantly in terms of both SHD and spurious rate. Notably, **GLIDE** achieves a remarkable 64.17% reduction rate on SHD over FCI. Furthermore, **GLIDE** incurs much less computational cost than both NOTEARS and MLP-NOTEARS (see the runtime plots) while achieving comparable or better performance. For example, **GLIDE** runs 96.54× and 15.66× faster than both NOTEARS and MLP-NOTEARS in in 100- and 500-node graphs while being second to MLP-NOTEARS in terms of SHD (with a small gap of 16.2%) and achieving best spurious rate in the largest graph setting with 500 nodes (4.2% vs the second best of 11.75% of NOTEARS and third best of 13.19% of MLP-NOTEARS). In the **nL-nG** cases, we also have a similar observations where **GLIDE** is again the second best in SHD and best in spurious rate (in the largest graph settings) while being much more scalable than the best and second best baselines in SHD and spurious rate, respectively. Our reported results also confirm our intuition earlier that score-based methods, despite being more scalable than constraint-based approaches, tend to perform much less robustly in large graph settings where the score function becomes less accurate. For example, both SCORE and DAS reach over 30% of spurious rate in 500-node graphs.

**Extreme Setting.** Figure 3 reports the performance of **GLIDE** in comparison to other baselines in this setting. In the **L-G** cases, it is observed that **GLIDE** achieves the best performance in both SHD and spurious rate while also achieving the fastest running time in most graph settings. In terms of SHD, the performance gaps between **GLIDE** and the second best (NOTEARS) and worst baselines (SCORE) are 11.74% and 108.5%, respectively. In terms of spurious rate, **GLIDE** also improves over the baselines with substantial performance gaps ranging from 5.46% and 48.61% against the second best and worst baselines, respectively. In the **nL-nG** cases, **GLIDE** and MLP-NOTEARS perform comparably as best baselines in SHD but **GLIDE** outperforms it by a gap of 8.23% in term of spurious rate. Furthermore, against the most high-performing baselines in this case, NOTEARS and MLP-NOTEARS, **GLIDE** achieve 9.6× and 25.52× faster processing time, respectively.

### 5.2 Real-World Data

The results show that **GLIDE** achieves the lowest SHD ($8.7 \pm 0.48$) and spurious rate (0%), significantly outperforms the second-best baseline, SCORE, whose SHD is $13.2 \pm 0.66$ and spurious rate of $22 \pm 4\%$. It is worth noting that **GLIDE** only takes $10.46 \pm 0.71$ seconds to accomplish this task, which is three times faster than SCORE (at $30.2 \pm 3.98$ seconds). FCI and DAS respectively stand at the third and forth place with the average SHD being 17 and $22.27 \pm 1.65$, and the corresponding spurious rates are 44% and $55 \pm 4\%$.

## 6 Conclusion

This paper presents a new perspective of causal learning via a new invariance test for causality that inspires a reliable and scalable algorithm for recovering causal graphs from observational data. Our approach explores a key insight that the effect-cause conditional distribution remains invariant under changes in the prior cause distribution, leading to a parent-finding procedure for each variable via synthetic data-augmentation. This procedure is further coupled with an effective search algorithm that exploits prior knowledge of each effect variable's Markov blanket and the sparsity of the causal graphs to significantly reduces the overall complexity. This, in turn, results in a significant reduction in complexity and marked improvements both in speed and accuracy over existing SOTA methods on various benchmark datasets.

## Impact Statement

Our work focuses on improving the scalability and reliability of the causal discovery procedure. The findings in this work offer a promising direction for further research and practical applications in causal inference, especially in scenarios involving large and complex datasets. As such, it could have significant broader impact on future research due to the ever-growing ubiquity of larger datasets. The source code and the datasets used for evaluation are publicly accessible, thus raising no ethical concerns.

## Acknowledgments

MS was supported by JST ASPIRE Grant Number JPMJAP2405. PLN was funded by Hanoi University of Science and Technology (HUST) under grant number T2024-TĐ-002. In addition, this work utilized GPU compute resource at SDSC and ACES through allocation CIS230391 from the Advanced Cyberinfrastructure Coordination Ecosystem: Services and Support (ACCESS) program (Boerner et al., 2023), which is supported by U.S. National Science Foundation grants #2138259, #2138286, #2138307, #2137603, and #2138296. The authors would also like to thank anonymous reviewers for their careful reviews and insightful comments.

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

# A  PROOFS

## A.1  Proof of Theorem 1

First, we establish the following **auxiliary result** which will be leveraged to establish our main result.

**Lemma 1.** $\forall \boldsymbol{Z} \subseteq \boldsymbol{X} \setminus \boldsymbol{B} : P_i(\boldsymbol{Z} \mid \boldsymbol{B}) = P(\boldsymbol{Z} \mid \boldsymbol{B})$. *Intuitively, this means the distribution of any set of non-source variables $\boldsymbol{Z}$ conditioned on source variables $\boldsymbol{B}$ remains invariant, i.e., $P_i(\boldsymbol{Z} \mid \boldsymbol{B}) = P(\boldsymbol{Z} \mid \boldsymbol{B})$, when we perturb the source marginals, i.e., replacing $P(\boldsymbol{B})$ with $P_i(\boldsymbol{B})$. As a direct consequence, we have $P_i(\boldsymbol{Z}) = \int_{\boldsymbol{B}} P_i(\boldsymbol{Z} \mid \boldsymbol{B}) P_i(\boldsymbol{B}) d\boldsymbol{B} = \int_{\boldsymbol{B}} P(\boldsymbol{Z} \mid \boldsymbol{B}) P_i(\boldsymbol{B}) d\boldsymbol{B}$.*

**Proof of Lemma 1**: We have $P(\boldsymbol{X}) = P(\boldsymbol{B}) P(\boldsymbol{X} \setminus \boldsymbol{B} \mid \boldsymbol{B})$ where $P(\boldsymbol{X} \setminus \boldsymbol{B} \mid \boldsymbol{B}) = \prod_{X \in \boldsymbol{X} \setminus \boldsymbol{B}} P(X \mid \mathrm{Pa}(X))$ is constant. Therefore: $P_i(\boldsymbol{Z} \mid \boldsymbol{B}) = \int_{\boldsymbol{X} \setminus (\boldsymbol{B} \cup \boldsymbol{Z})} P_i(\boldsymbol{X} \setminus \boldsymbol{B} \mid \boldsymbol{B}) d\boldsymbol{B} = \int_{\boldsymbol{X} \setminus (\boldsymbol{B} \cup \boldsymbol{Z})} P(\boldsymbol{X} \setminus \boldsymbol{B} \mid \boldsymbol{B}) d\boldsymbol{B} = P(\boldsymbol{Z} \mid \boldsymbol{B})$.  $\square$

**Key result:** For each node $X$, its candidate parent set is therefore a power set of $\mathrm{Pa}(X) \cup \mathrm{Ch}(X)$ (see Theorem 7). Our key result then establishes that over the random choice $P_i(B)$ as prior over sources $B$: (1) the variance over the induced conditional $P_i(X \mid \mathrm{Pa(X)})$ is zero, $\mathbb{V}_{P_i(\boldsymbol{B})}[P_i(X \mid \mathrm{Pa}(X))] = 0$; and (2) the variance over $P_i(X \mid \boldsymbol{Z})$ where $\boldsymbol{Z} \subseteq \mathrm{Pa}(X) \cup \mathrm{Ch}(X)$ and $\boldsymbol{Z} \neq \mathrm{Pa}(X)$ is not zero. This means we can use $\mathbb{V}_{P_i(\boldsymbol{B})}[P_i(X \mid \boldsymbol{Z})]$ where $\boldsymbol{Z} \subseteq \mathrm{Pa}(X) \cup \mathrm{Ch}(X)$ to determine the true causal parents of $X$. The proof is as following:

Since we only perform our invariance test on non-source variables $X$, we consider the following tuple $(X, \boldsymbol{Z})$ where $\boldsymbol{Z} = \boldsymbol{Z}_1 \cup \boldsymbol{Z}_2$, $\boldsymbol{Z}_1 \cap \boldsymbol{Z}_2 = \emptyset$, $\boldsymbol{Z}_1 \cap \boldsymbol{B} = \emptyset$, and $\boldsymbol{Z}_2 \subseteq \boldsymbol{B}$. Then we have:

$$P_i(X \mid \boldsymbol{Z}) = P_i(X \mid \boldsymbol{Z}_1, \boldsymbol{Z}_2) = \frac{P_i(X, \boldsymbol{Z}_1, \boldsymbol{Z}_2)}{P_i(\boldsymbol{Z}_1, \boldsymbol{Z}_2)} = \frac{\int_{\boldsymbol{B} \setminus \boldsymbol{Z}_2} P(X, \boldsymbol{Z}_1 \mid \boldsymbol{B}) P_i(\boldsymbol{B}) d(\boldsymbol{B} \setminus \boldsymbol{Z}_2)}{\int_{\boldsymbol{B} \setminus \boldsymbol{Z}_2} P(\boldsymbol{Z}_1 \mid \boldsymbol{B}) P_i(\boldsymbol{B}) d(\boldsymbol{B} \setminus \boldsymbol{Z}_2)} \tag{6}$$

Here, the last equality follows from Lemma 1. This means if $\mathbb{V}_{P_i(\boldsymbol{B})}[P_i(X \mid \boldsymbol{Z})] = 0$, then $\frac{P_i(X, \boldsymbol{Z}_1, \boldsymbol{Z}_2)}{P_i(\boldsymbol{Z}_1, \boldsymbol{Z}_2)}$ must be a constant function $\alpha(X, \boldsymbol{Z}_1, \boldsymbol{Z}_2)$ with respect to $P_i(\boldsymbol{B})$. Given its expansion above, we must have:

$$P_i(X, \boldsymbol{Z}_1 \mid \boldsymbol{B}) = \alpha(X, \boldsymbol{Z}) P_i(\boldsymbol{Z}_1 \mid \boldsymbol{B}) \Rightarrow P_i(X \mid \boldsymbol{Z}_1, \boldsymbol{B}) = P_i(X \mid \boldsymbol{Z}_1, \boldsymbol{Z}_2, \boldsymbol{B} \setminus \boldsymbol{Z}_2) = \alpha(X, \boldsymbol{Z}_1, \boldsymbol{Z}_2), \tag{7}$$

where $\alpha(X, \boldsymbol{Z}_1, \boldsymbol{Z}_2)$ is a scalar function independent of the choice $P_i(\boldsymbol{B})$ for source prior. Equivalently, this means knowing $\boldsymbol{B} \setminus \boldsymbol{Z}_2$ does not change the condition of $X$ on $\boldsymbol{Z} = (\boldsymbol{Z}_1, \boldsymbol{Z}_2)$. Following the definition of conditional independence, we have $X \perp (\boldsymbol{B} \setminus \boldsymbol{Z}_2) \mid (\boldsymbol{Z}_1, \boldsymbol{Z}_2)$. Hence,

$$\mathbb{V}_{P_i(\boldsymbol{B})}[P_i(X \mid \boldsymbol{Z})] = 0 \Rightarrow X \perp (\boldsymbol{B} \setminus \boldsymbol{Z}_2) \mid (\boldsymbol{Z}_1, \boldsymbol{Z}_2) \tag{8}$$

On the other hand, if $X \perp (\boldsymbol{B} \setminus \boldsymbol{Z}_2) \mid \boldsymbol{Z}_1, \boldsymbol{Z}_2$ holds, $P(X \mid (\boldsymbol{B} \setminus \boldsymbol{Z}_2), \boldsymbol{Z}_1, \boldsymbol{Z}_2) = P(X \mid \boldsymbol{Z}_1, \boldsymbol{Z}_2)$. Plugging this into Equation 6 results in:

$$
\begin{aligned}
P_i(X \mid \boldsymbol{Z}) \quad &= \quad P_i(X \mid \boldsymbol{Z}_1, \boldsymbol{Z}_2) \\
&= \quad \frac{\int_{\boldsymbol{B} \setminus \boldsymbol{Z}_2} P(X, \boldsymbol{Z}_1 \mid \boldsymbol{B}) P_i(\boldsymbol{B}) d(\boldsymbol{B} \setminus \boldsymbol{Z}_2)}{\int_{\boldsymbol{B} \setminus \boldsymbol{Z}_2} P(\boldsymbol{Z}_1 \mid \boldsymbol{B}) P_i(\boldsymbol{B}) d(\boldsymbol{B} \setminus \boldsymbol{Z}_2)} \qquad \triangleleft \text{Apply Lemma 1 for } P(X, \boldsymbol{Z}_1 \mid \boldsymbol{B}) \\
&= \quad \frac{\int_{\boldsymbol{B} \setminus \boldsymbol{Z}_2} P(X \mid \boldsymbol{Z}_1, \boldsymbol{Z}_2, \boldsymbol{B} \setminus \boldsymbol{Z}_2) P(\boldsymbol{Z}_1 \mid \boldsymbol{B}) P_i(\boldsymbol{B}) d(\boldsymbol{B} \setminus \boldsymbol{Z}_2)}{\int_{\boldsymbol{B} \setminus \boldsymbol{Z}_2} P(\boldsymbol{Z}_1 \mid \boldsymbol{B}) P_i(\boldsymbol{B}) d(\boldsymbol{B} \setminus \boldsymbol{Z}_2)} \\
&= \quad \frac{\int_{\boldsymbol{B} \setminus \boldsymbol{Z}_2} P(X \mid \boldsymbol{Z}_1, \boldsymbol{Z}_2) P(\boldsymbol{Z}_1 \mid \boldsymbol{B}) P_i(\boldsymbol{B}) d(\boldsymbol{B} \setminus \boldsymbol{Z}_2)}{\int_{\boldsymbol{B} \setminus \boldsymbol{Z}_2} P(\boldsymbol{Z}_1 \mid \boldsymbol{B}) P_i(\boldsymbol{B}) d(\boldsymbol{B} \setminus \boldsymbol{Z}_2)} \qquad \triangleleft \text{Conditional independence holds} \\
&= \quad P(X \mid \boldsymbol{Z}_1, \boldsymbol{Z}_2) \frac{\int_{\boldsymbol{B} \setminus \boldsymbol{Z}_2} P(\boldsymbol{Z}_1 \mid \boldsymbol{B}) P_i(\boldsymbol{B}) d(\boldsymbol{B} \setminus \boldsymbol{Z}_2)}{\int_{\boldsymbol{B} \setminus \boldsymbol{Z}_2} P(\boldsymbol{Z}_1 \mid \boldsymbol{B}) P_i(\boldsymbol{B}) d(\boldsymbol{B} \setminus \boldsymbol{Z}_2)} \\
&= \quad P(X \mid \boldsymbol{Z}_1, \boldsymbol{Z}_2)
\end{aligned}
$$

which does not depend on $P_i(\boldsymbol{B})$. This means:

$$X \perp (\boldsymbol{B} \setminus \boldsymbol{Z}_2) \mid (\boldsymbol{Z}_1, \boldsymbol{Z}_2) \Rightarrow \mathbb{V}_{P_i(\boldsymbol{B})}[P(X \mid \boldsymbol{Z})] = 0. \tag{9}$$

Combining Equations 8 and 9, we obtain a **bidirectional equivalence**:

$$\mathbb{V}_{P_i(\boldsymbol{B})}[P_i(X \mid \boldsymbol{Z}_1, \boldsymbol{Z}_2)] = 0 \Leftrightarrow X \perp (\boldsymbol{B} \setminus \boldsymbol{Z}_2) \mid \boldsymbol{Z}_1, \boldsymbol{Z}_2. \tag{10}$$

Assuming that $\mathrm{Pa}(X) \cap \mathrm{Spouse}(X) = \emptyset$ and $\mathrm{Spouse}(X) \cap \boldsymbol{B} = \emptyset$, the above **bidirectional equivalence** in Equation 10 has these direct consequences:

a. $\boldsymbol{Z}_1 \cup \boldsymbol{Z}_2 = \mathrm{Pa}(X) \Rightarrow X \perp (\boldsymbol{B} \setminus \boldsymbol{Z}_2) \mid (\boldsymbol{Z}_1 \cup \boldsymbol{Z}_2) \Rightarrow \mathbb{V}_{P_i(\boldsymbol{B})}[P_i(X \mid \boldsymbol{Z})] = 0$

b. $\boldsymbol{Z}_1 \cup \boldsymbol{Z}_2 = \mathrm{Pa}(X) \cup \{Y\}$ where $Y \in \mathrm{Ch}(X)$, then when conditioned on $Y$, there exists an active (backdoor) path $\boldsymbol{B} \rightarrow \cdots \rightarrow \mathrm{Spouse}(X) \rightarrow Y \leftarrow X$. Therefore, $X \perp (\boldsymbol{B} \setminus \boldsymbol{Z}_2) \mid (\boldsymbol{Z}_1 \cup \boldsymbol{Z}_2)$ does not hold. Hence, $\mathbb{V}_{P_i(\boldsymbol{B})}[P_i(X \mid \boldsymbol{Z})] > 0$ because otherwise, $\mathbb{V}_{P_i(\boldsymbol{B})}[P_i(X \mid \boldsymbol{Z})] = 0$ implies $X \perp (\boldsymbol{B} \setminus \boldsymbol{Z}_2) \mid (\boldsymbol{Z}_1 \cup \boldsymbol{Z}_2)$ which contradicts the above backdoor consequence.

c. $\mathrm{Pa}(X) \not\subseteq \boldsymbol{Z}_1 \cup \boldsymbol{Z}_2$ also implies the existence of an active backdoor connecting $\boldsymbol{B}$ and $X$. Using the same argument as in (b), we know this case also implies $\mathbb{V}_{P_i(\boldsymbol{B})}[P_i(X \mid \boldsymbol{Z})] > 0$.

Overall, (b) and (c) collectively ensure that as long as $\boldsymbol{Z}$ is not $\mathrm{Pa}(X)$, $\mathbb{V}_{P_i(\boldsymbol{B})}[P_i(X \mid \boldsymbol{Z})] > 0$. Otherwise, (a) ensures that $\mathbb{V}_{P_i(\boldsymbol{B})}[P_i(X \mid \boldsymbol{Z})] = 0$. $\qquad\square$

We also note that the structural assumptions $\mathrm{Pa}(X) \cap \mathrm{Spouse}(X) = \emptyset$ and $\mathrm{Spouse}(X) \cap \boldsymbol{B} = \emptyset$ hold with relatively high probability. For example, regarding real-world graphs (provided by bnlearn (Scutari, 2009)) we presented, the percentage of nodes satisfying these assumptions is over 75%.

## A.2 Proof of Theorem 2

Let $\boldsymbol{S}(G) \subseteq \boldsymbol{X}$ and $\boldsymbol{B}(G) \subseteq \boldsymbol{X}$ denote respectively the set of sources and an arbitrary basis set of the DAG $G = (\boldsymbol{X}, \boldsymbol{E})$. We will prove the following inductive statement:

$P(n)$: *"Given a DAG $G$ of $n$ $(n \geq 1)$ nodes, we have $|\boldsymbol{B}(G)| \leq |\boldsymbol{S}(G)|$"*

To see this, note that in the base case $n = 1$, $G$ has single node which is obviously both a source and basis node. Hence, $|\boldsymbol{B}(G)| = |\boldsymbol{S}(G)|$ and $P(1)$ is true.

Now, suppose $P(n)$ is true, we will complete induction by showing that $P(n + 1)$ is also true. To show this, let $X$ be a terminal node in $G$ that has no children. Removing $X$ from $G$ thus results in another DAG $G'$ with $n$ nodes.

Let $\boldsymbol{B}(G')$ denote an arbitrary basis set of $G'$. By definition, nodes in $\boldsymbol{B}(G')$ are mutually independent and for any nodes $X' \notin \boldsymbol{X} \setminus (\boldsymbol{B}(G') \cup \{X\})$, there exists a node $Z$ in $\boldsymbol{B}(G')$ such that $X' \not\perp Z$ following the definition of $d$-separation.

Choosing $X' \in \mathrm{Pa}[X]$, there must exist $Z \in \boldsymbol{B}(G')$ such that $Z$ is connected to $X'$ via $d$-separation. Since $X'$ is a parent of $X$, $Z$ is also connected to $X$ via $d$-separation. This means each node in $G$ is either a part of $\boldsymbol{B}(G')$ or connected to another node in $\boldsymbol{B}(G')$ via $d$-separation. Hence, $\boldsymbol{B}(G')$ is also a basis set of $G$. In this case, $|\boldsymbol{B}(G')| = |\boldsymbol{S}(G')| \leq |\boldsymbol{S}(G)|$ since $P(n)$ is true.

Otherwise, suppose $\boldsymbol{B}(G)$ is a basis set of $G$ that is not in $G'$. In this case, $\boldsymbol{B}(G)$ must contain $X$ and by definition of basis (see Definition 3) and its choice, $X$ must be an isolated node with no parents and children. This also means $\boldsymbol{B}(G) \setminus \{X\}$ is a basis set of $G'$. Otherwise, there exists a node in $G$ that is not connected to any nodes in the basis set $\boldsymbol{B}(G)$, resulting in a contradiction. As such, we have $|\boldsymbol{B}(G)| = |\boldsymbol{B}(G) \setminus \{X\}| + 1 \leq |\boldsymbol{S}(G')| + 1 = |\boldsymbol{S}(G)|$ where the first inequality is true due to $P(n)$ and the last equality is true since $X$ is an isolated node which is also a source node.

As such, $|\boldsymbol{B}(G)| \leq |\boldsymbol{S}(G)|$ meaning $P(n + 1)$ is true if $P(n)$ is true. Induction completes. $\qquad\square$

It is worth noting that for every graph $G$, there exists a basis set $\boldsymbol{B}$ such that $|\boldsymbol{B}(G)| = |\boldsymbol{S}(G)|$. The trivial solution is $\boldsymbol{B} \equiv \boldsymbol{S}$ since the set of sources $\boldsymbol{S}$ satisfies Definition 3 of a basis set (i.e., source variables are mutually independent and every other variable is dependent on at least one source).

### A.3 Proof of Theorem 3

To prove that the procedure in Theorem 3 produces a basis set with maximum size, it suffices to prove that each step in this procedure removes exactly one source from the graph (see Lemma 2). In this case, the returned basis set has the same size as the source set, which is also the maximum basis set according to Theorem 2.

**Lemma 2.** *If $X$ has the smallest number of dependent nodes, i.e., $X \triangleq \arg\min_{Y \in \boldsymbol{V}} |\Phi(Y)|$, then $X$ is dependent on exactly one source in $G = (\boldsymbol{V}, \boldsymbol{E})$.*

**Proof of Lemma 2:** Let statement $(P)$ be "$X \in \boldsymbol{V}$ has the smallest number of dependent nodes in $G$", and $(Q)$ be "$X$ is dependent on exactly one source in $G$". To prove $(P) \Rightarrow (Q)$, we will prove that its negation $(P) \wedge \neg(Q)$ is false. Specifically, $(P) \wedge \neg(Q)$ reads "$X$ has the smallest number of dependent nodes and $X$ is dependent on two or more sources in $G$".

To see this, when $\neg(Q)$ is true, there must exist two source variables $S_1$ and $S_2$ which are dependent on $X$ (note that $S_1$ and $S_2$ are independent by definition). This means there must exist an active path from $S_1$ to $X$, and another from $S_2$ to $X$. Consequently, $\forall Y \in \Phi(S_1)$, there exists an active path from $S_1$ to $Y$ and hence, $X$ and $Y$ are connected by an active path via a common cause trail at $S_1$ which implies $Y \in \Phi(X)$. Likewise, $Y \in \Phi(S_2)$ also implies $Y \in \Phi(X)$. This means $|\Phi(X)| \geq |\Phi(S_1) \cup \Phi(S_2)|$. Finally, note that (1) $S_2 \notin \Phi(S_1)$ as both are sources; and (2) $S_2 \in \Phi(S_2)$ by definition since $S_2 \not\perp\!\!\!\perp S_2$. Given this, $|\Phi(S_1) \cup \Phi(S_2)| \geq |\Phi(S_1)| + 1 > |\Phi(S_1)|$. Combining this with the earlier established fact that $|\Phi(X)| \geq |\Phi(S_1) \cup \Phi(S_2)|$ leads to $|\Phi(X)| > |\Phi(S_1)|$ which is a contradiction to $(P)$. Thus $(P) \wedge \neg(Q)$ is false. $\square$

### A.4 Proof of Theorem 4

Let $D_i$ be any downsampled dataset of $D$. Consider a candidate value $\boldsymbol{b}$ of ~~the source~~ a variable $\boldsymbol{B}$ that occurs $D$. Let $n(\boldsymbol{b})$ and $n_i(\boldsymbol{b})$ denote the corresponding numbers of data points in $D$ and $D_i$ that have $\boldsymbol{B} = \boldsymbol{b}$. Since $D_i$ is downsampled from $D$, $n_i(\boldsymbol{b}) \leq n(\boldsymbol{b})$. Hence, let $n \triangleq |D|$ and $n_i \triangleq |D_i|$,

$$
\begin{aligned}
P(\boldsymbol{B} = \boldsymbol{b})/P_i(\boldsymbol{B} = \boldsymbol{b}) &\geq (n(\boldsymbol{b})/n)/(n_i(\boldsymbol{b})/n_i) \\
&= (n(\boldsymbol{b})/n_i(\boldsymbol{b})) \cdot (n_i/n) \geq n_i/n = |D_i|/|D|
\end{aligned}
\tag{11}
$$

which follows from the facts that $P(\boldsymbol{B} = \boldsymbol{b}) = n(\boldsymbol{b})/n$, $P_i(\boldsymbol{B} = \boldsymbol{b}) = n_i(\boldsymbol{b})/n_i$, and $n(\boldsymbol{b}) \geq n_i(\boldsymbol{b})$ due to the downsampled nature of $D_i$. Taking the minimum over all candidate values $\boldsymbol{b}$ of $\boldsymbol{B}$,

$$
\gamma_i \triangleq \min_{\boldsymbol{b}} \left( P(\boldsymbol{B} = \boldsymbol{b})/P_i(\boldsymbol{B} = \boldsymbol{b}) \right) \geq |D_i|/|D|
\tag{12}
$$

As a result, for all downsampled dataset $D_i$ of $D$, the downsampling rate $|D|/|D_i| \geq \gamma_i$ where $\gamma_i$ is defined in terms of the original and target marginal over the source variable $\boldsymbol{B}$ as stated in Eq. (12). Hence, if $D_i$ is the downsampled data with minimum downsampling rate $|D|/|D_i| \geq \gamma_i^{-1}$.

### A.5 Proof of Theorem 5

Since $D_i$ is created via sampling with no replacement $P_i(\boldsymbol{B} = \boldsymbol{b}) \cdot |D| \cdot \gamma_i$ data points from $D$ where $\boldsymbol{B} = \boldsymbol{b}$, we know that $n_i(\boldsymbol{b}) = P_i(\boldsymbol{B} = \boldsymbol{b}) \cdot |D| \cdot \gamma_i = P_i(\boldsymbol{B} = \boldsymbol{b}) \cdot n \cdot \gamma_i$. Thus,

$$
\begin{aligned}
n_i &\triangleq \left( \sum_{\boldsymbol{b}} n_i(\boldsymbol{b}) \right) = \left( \sum_{\boldsymbol{b}} P_i(\boldsymbol{B} = \boldsymbol{b}) \cdot |D| \cdot \gamma_i \right) \\
&= P_i(\boldsymbol{B} = \boldsymbol{b}) \cdot n \cdot \gamma_i = n \cdot \gamma_i \cdot \left( \sum_{\boldsymbol{b}} P_i(\boldsymbol{B} = \boldsymbol{b}) \right) = n \cdot \gamma_i .
\end{aligned}
\tag{13}
$$

As a result, $|D_i|/|D| = n_i/n = \gamma_i$ and $n_i(\boldsymbol{b})/n_i = (P_i(\boldsymbol{B} = \boldsymbol{b}) \cdot n \cdot \gamma_i)/(n \cdot \gamma_i) = P_i(\boldsymbol{b})$. Thus, $D_i \sim P_i(\boldsymbol{X})$ and the downsampling rate $|D|/|D_i|$ achieves minimum as expected.

### A.6 Proof of Theorem 6

**A. Convexity.** We will first prove that the subspace of $P_i(\boldsymbol{B})$ that satisfies $\gamma_i \geq \gamma_o$ is convex. To see this, consider $P_i^1(\boldsymbol{B})$ and $P_i^2(\boldsymbol{B})$ whose (inverse) downsampling rates (see Theorem 4) $\gamma_i^1$ and $\gamma_i^2$ are both larger than $\gamma_o$. This means both $P_i^1(\boldsymbol{B})$ and $P_i^2(\boldsymbol{B})$ belong to the aforementioned subspace.

Now, let $\alpha \in (0,1)$ and $P_i^\alpha(\boldsymbol{B}) = \alpha \cdot P_i^1(\boldsymbol{B}) + (1-\alpha) \cdot P_i^2(\boldsymbol{B})$. The (inverse) downsampling rate of $P_i^\alpha(\boldsymbol{B})$ is defined as

$$\gamma_i^\alpha \triangleq \min_{\boldsymbol{b}} \left( \frac{P(\boldsymbol{B}=\boldsymbol{b})}{P_i(\boldsymbol{B}=\boldsymbol{b})} \right) = \min_{\boldsymbol{b}} \left( \frac{P(\boldsymbol{B}=\boldsymbol{b})}{\alpha \cdot P_i^1(\boldsymbol{B}=\boldsymbol{b}) + (1-\alpha) \cdot P_i^2(\boldsymbol{B}=\boldsymbol{b})} \right) \tag{14}$$

$$\geq \min_{\boldsymbol{b}} \left( \frac{P(\boldsymbol{B}=\boldsymbol{b})}{\alpha \cdot \gamma_o^{-1} \cdot P(\boldsymbol{B}=\boldsymbol{b}) + (1-\alpha) \cdot \gamma_o^{-1} \cdot P(\boldsymbol{B}=\boldsymbol{b})} \right) = \frac{1}{\gamma_o^{-1}} = \gamma_o , \tag{15}$$

where the inequality follows from the facts that (1) $P(\boldsymbol{B}=\boldsymbol{b})/P_i^1(\boldsymbol{B}=\boldsymbol{b}) \geq \gamma_i^1 \geq \gamma_o$; and $P(\boldsymbol{B}=\boldsymbol{b})/P_i^1(\boldsymbol{B}=\boldsymbol{b}) \geq \gamma_i^2 \geq \gamma_o$ which follows from the (inverse) downsampling rate's definition in Theorem 4. This means $P_i^1(\boldsymbol{B}=\boldsymbol{b}) \leq \gamma_o^{-1} \cdot P(\boldsymbol{B}=\boldsymbol{b})$ and $P_i^2(\boldsymbol{B}=\boldsymbol{b}) \leq \gamma_o^{-1} \cdot P(\boldsymbol{B}=\boldsymbol{b})$, which can plugged into Eq. (14) to arrive at Eq. (15). This in turn implies $P_i^\alpha(\boldsymbol{B})$ belongs to the subspace of source priors with the induced (inverse) downsampling rate above $\gamma_o$. Thus, following the definition of convexity, we know that this subspace is convex.

**B. Convex Hull.** Note that any source prior $P_i(\boldsymbol{B})$ is a point in an $r$-dimensional simplex where $r$ is the number of candidate values of the source variables $\boldsymbol{B}$. The convex hull of the aforementioned convex set can then be determined by finding its intersection with the edges of the simplex. This comprises $r$ points of the following form:

$$P^{(k)}(\boldsymbol{B}) = \alpha_k \cdot P(\boldsymbol{B}) + (1-\alpha_k) \cdot \delta_k(\boldsymbol{B}) , \tag{16}$$

where $\alpha_k$ is selected such that $\gamma_i^{(k)} \triangleq \min_{\boldsymbol{b}}(P(\boldsymbol{B}=\boldsymbol{b})/P^{(k)}(\boldsymbol{B}=\boldsymbol{b})) = \gamma_o$. Solving for $\alpha_k$ results in the following equation:

$$\gamma_o = \min \left( \frac{1}{\alpha_k}, \frac{P\left(\boldsymbol{B}=\boldsymbol{b}^{(k)}\right)}{\alpha_k \cdot P\left(\boldsymbol{B}=\boldsymbol{b}^{(k)}\right) + \left(1-\alpha_k\right)} \right) = \frac{P\left(\boldsymbol{B}=\boldsymbol{b}^{(k)}\right)}{\alpha_k \cdot P\left(\boldsymbol{B}=\boldsymbol{b}^{(k)}\right) + \left(1-\alpha_k\right)} , \tag{17}$$

where $\boldsymbol{b}^{(k)}$ denote the $k$-th candidate value of $\boldsymbol{B}$. The second equality in the above holds since $\alpha_k \in (0,1)$ which means $\alpha_k \cdot P(\boldsymbol{B}=\boldsymbol{b}) \leq \alpha_k \cdot P(\boldsymbol{B}=\boldsymbol{b}) + 1 - \alpha_k$ or equivalently, $1/\alpha_k \geq P(\boldsymbol{B}=\boldsymbol{b})/(\alpha_k \cdot P(\boldsymbol{B}=\boldsymbol{b}) + 1 - \alpha_k)$. This allows us to get rid of the min operator in Eq. (17) and consequently, compute $\alpha_k$ in closed-form:

$$\alpha_k = \left( 1 - \gamma_o^{-1} \cdot P\left(\boldsymbol{B}=\boldsymbol{b}^{(k)}\right) \right) / \left( 1 - P\left(\boldsymbol{B}=\boldsymbol{b}^{(k)}\right) \right) . \tag{18}$$

This completes our derivation for the convex hull in Theorem 6. $\qquad\square$

### A.7 Proof of Theorem 7

Suppose $U, V \in \mathrm{Pa}[X]$, it must follow that $U \in \mathbb{M}(V)$ and $V \in \mathbb{M}(U)$ since $U$ and $V$ are spouses. This means $(U, V) \in \boldsymbol{E}$. Hence, there is a bidirectional edge in $G'(X)$ between any two parents of $X$ which means there exists at least a clique in $G'(X)$ containing $\mathrm{Pa}[X]$. $\qquad\square$

## B ALGORITHMS & ANALYSIS

### B.1 Pseudo-code for GLIDE

### B.2 Find the Basis of a graph

This section provides visualization and pseudocode of the basis finding procedure in Theorem 3.

---

**Algorithm 1 GLIDE**

---

**Input:** Data $D$, the set of all variables $\boldsymbol{X}$, down-sampling ratio threshold $\gamma$, number of down-sampling sub-datasets $m$.
**Output:** DAG $G$.

1: Find the basis set $\boldsymbol{B}$ using Algorithm 2.
2: Compute $m$ priors $P_i(\boldsymbol{B})$ by Theorem 6.
3: Sample $m$ sub-datasets $\{D_i \sim P_i(\boldsymbol{B})\}$ from $D$ (Section 4.2.2).
4: Use the GSMB algorithm to recover the Markov blanket for each node $\mathbb{M}(X), \forall X \in \boldsymbol{X}$.
5: Find the plausible parent set $\mathcal{Z}$ for each variable $X$ using Algorithm 3.
6: Find causal parent $\text{Pa}[X]$ for each $X \in \boldsymbol{X}$ by Theorem 1 using $\{D_i\}_{i=1}^m$, its plausible parent set $\mathcal{Z}$.
7: Build $G$ using $(X, \text{Pa}[X])$ for each $X \in \boldsymbol{X}$.
8: **Return** $G$.

---

---

**Algorithm 2** Maximum-Sized Basis Search

---

**Input:** Data $D$ and the set of all variable indices $\boldsymbol{V}$.
**Output:** Maximum-Sized Basis $\boldsymbol{B}$.

1: Compute $[\boldsymbol{\Phi}]_{ij} = \mathbb{I}(X_i \not\perp\!\!\!\perp X_j)$ using existing statistical independence test.
2: **while** $|\boldsymbol{V}| > 0$ **do**
3:     Choose $t = \arg\min_{i \in \boldsymbol{V}} \sum_j [\boldsymbol{\Phi}]_{ij}$
4:     Update $\boldsymbol{B} = \boldsymbol{B} \cup \{X_t\}$
5:     Update $\boldsymbol{V} \leftarrow \boldsymbol{V} \setminus \{i : [\boldsymbol{\Phi}]_{ti} = 1\}$
6: **end while**
7: **Return** $\boldsymbol{B}$

---

**A. Pseudocode.** First, the pseudocode of the basis finding algorithm is detailed below.

This algorithm is guaranteed to find the maximum-sized of basis variables since each of its iteration will remove exactly one source node from $\boldsymbol{V}$ as proved in Appendix A.3. Thus, the number of nodes added to the basis is also the number of source nodes. Such basis is guaranteed to has maximum size following Theorem 2. A visualization of this algorithm is provided in Figure 4.

**B. Time Complexity.** Algorithm 2 consists of (i) computing the dependence network matrix $\boldsymbol{\Phi}$, and (ii) selecting the node with minimum number of dependents from $\boldsymbol{V}$ per iteration. The complexity for step (i) is $O(d^2)$ WHILE the complexity for step (ii) is $O(d)$ per iteration. As there are at most $d$ iterations, the total cost of Algorithm 2 is therefore $O(d^2)$.

### B.3 Find Plausible Parent Sets

Theorem 7 implies that if two nodes in the Markov blanket of $X$ do not belong to each other's Markov blanket, then they can not both be parents of $X$. According to this logic, we wish to find all sets of nodes whose Markov blanket contains one another. To do this, we develop a tree-based recursive algorithm that works as follows. Given the Markov blanket of all nodes, we perform Algorithm 3, which is a modification of the Bron-Kerbosch algorithm (Bron & Kerbosch, 1973). Instead of running Bron-Kerbosch for each node $X$ and its Markov blanket $\mathbb{M}(X)$, we add a virtual node that directly connects to all other nodes in the graph and start building the set of plausible parent sets for all nodes recursively from this virtual node.

Note that Algorithm 3 does not create a sub-tree (branch) if the search space of the root of that sub-tree is guaranteed to re-produce the content of a previously visited leaf and generate no new information – see lines 4-7. Otherwise, a plausible parent set of $X$ is the set of variables in the path from the search node containing $X$ to a leaf node, which is computed following the recursion of Algorithm 3. A step-by-step visualization of this algorithm is illustrated in Figure 5.

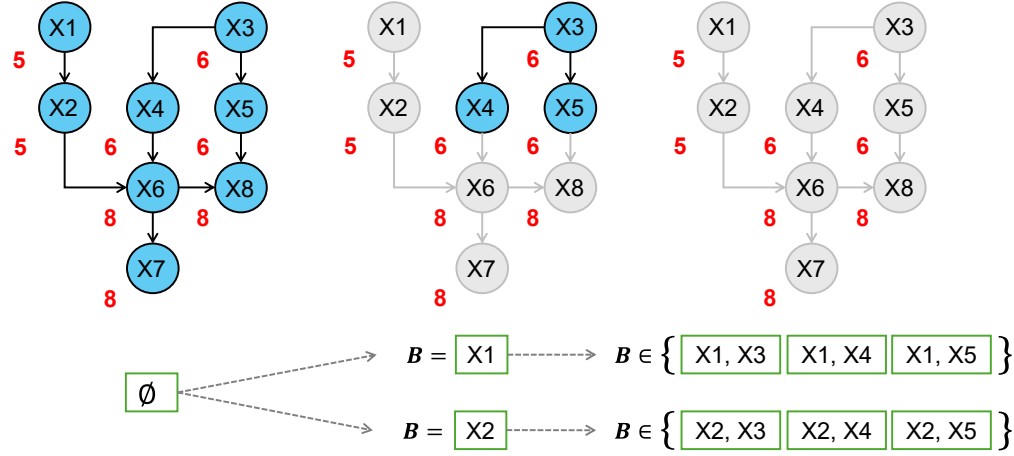

Figure 4: A step-by-step visualization of Algorithm 2. From left to right: each step finds the node with minimum number of dependents and remove it along with its dependent set from the graph. Each node is accompanied by a number in red that indicates the number of variables being dependent on that node. The removed nodes are colored gray WHILE active nodes are blue. The example graph is taken from the ASIA dataset (Scutari, 2009).

---

**Algorithm 3** Finding Plausible Parent Sets

**Inputs:** root node $r_o$ containing the virtual variable $X_o$, Markov blanket $\mathbb{M}(X)$ of all variable $X$.
**Outputs:** a plausible-parent-set tree, the path from each intermediate search node of variable $X$ to each leaf node represents a plausible candidate for Pa[X]

1: Initialize an empty list of leaves $\boldsymbol{L} \leftarrow \emptyset$;
2: search$(r_o) \leftarrow$ the set of all variables $\boldsymbol{X}$;
3: **Procedure** Recursive-build$(r)$
4: **for** $\ell$ in $\boldsymbol{L}$ **do**
5:    **if** path$(r_o \to r) \cup$ search$(r) \in$ path$(r_o \to \ell)$ **then**
6:       **Return**
7:    **end if**
8: **end for**
9: $\boldsymbol{S} \leftarrow$ sort $X_i \in$ search$(r)$ in the decreasing order of $|\text{search}(r) \cap \mathbb{M}(X_i)|$
10: $\boldsymbol{V} \leftarrow \emptyset$     #initialize the set of visited nodes
11: **if** $|\boldsymbol{S}| > 0$ **then**
12:    **while** $|\boldsymbol{S}| > 0$ **do**
13:       $X \leftarrow \boldsymbol{S}$.pop()
14:       $q \leftarrow$ node$(X)$
15:       search$(q) \leftarrow (\mathbb{M}(X) \cap \text{search}(r)) \setminus \boldsymbol{V}$
16:       path$(q) \leftarrow$ path$(r)) \cup \{X\}$
17:       Recursive-Build$(q)$
18:       $\boldsymbol{V} \leftarrow \boldsymbol{V} \cup \{X\}$
19:    **end while**
20: **else**
21:    $\boldsymbol{L} \leftarrow \boldsymbol{L} \cup \{r\}$
22: **end if**
23: **End Procedure**

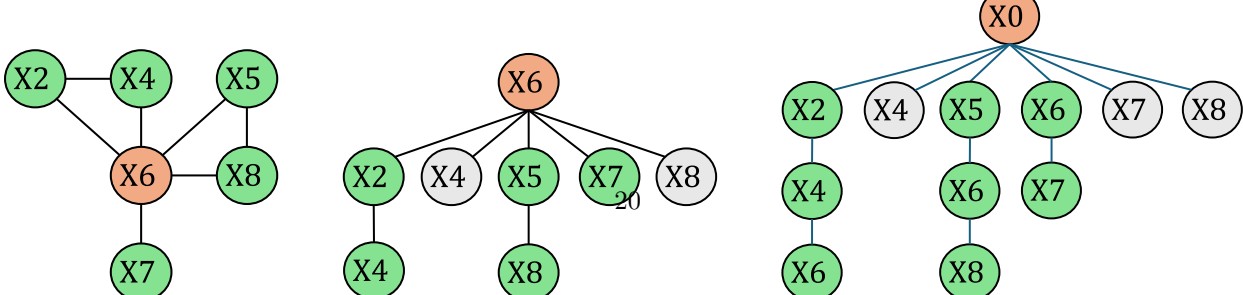

Table 1: **Baselines and Data Models.** The table indicates the applicability of each baseline to each data models. Applicability is determined based on the baseline's empirical effectiveness on recovering the causal graph from observational data under the corresponding data model. Note that except for our versatile method, most other baselines are not applicable to all data models (i.e., producing very poor performance that is not meaningful for comparison).

| Data Models | PC | GIES | FCI | Notears | MLP-Notears | DAS | SCORE | **GLIDE**(Ours) |
|---|---|---|---|---|---|---|---|---|
| Linear, Gaussian (**L-G**) | | ✓ | ✓ | ✓ | ✓ | ✓ | ✓ | ✓ |
| Non-linear, non-Gaussian (**nL-nG**) | | ✓ | ✓ | ✓ | ✓ | ✓ | ✓ | ✓ |
| Categorical, Synthetic | ✓ | ✓ | | | | | | ✓ |
| Categorical, Real-world | ✓ | ✓ | | | | | | ✓ |

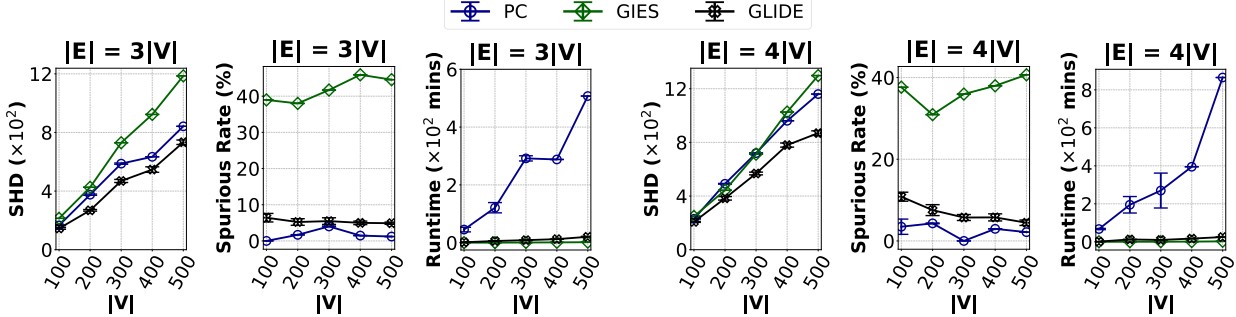

Figure 6: Reported performance of the baselines across two cases: (1) $\mathbf{E} = 3\mathbf{V}$ and (2) $\mathbf{E} = 4\mathbf{V}$

# C  ADDITIONAL EXPERIMENTS

## C.1  Experiment settings

**Baselines & Hyper-parameters for baselines.** In this section, we present the experimental settings for each baseline used in the empirical report in the Section V in the manuscript and Appendix sections C.2, C.3. Firstly, the code for PC is from the Bnlearn package(Scutari, 2009), provided in Python. We reported results of the PC-stable, which is an upgraded version of PC. On the other hand, the implementation of GIES and FCI are provided from the cdt package (Kalainathan et al., 2020) and the causal-learn package (Zheng et al., 2024), respectively. These algorithms require no tuned parameters as input and therefore can be used as recommended from the package from which they are provided. Notears and MLP-Notears are open-source on Github, and are provided by Zheng et al. (2018). For our experiments, we reuse the parameters recommendation in the original papers: Notears ($\lambda_1 = 0.1, w_{th} = 0.3$) and MLP-Notears ($\lambda_1 = \lambda_2 = 0.01, w_{th} = 0.3$). It is worth noting that we do not modify the original architecture of the network used in MLP-Notears. Lastly, DAS and SCORE are also open-source and are available from (Montagna et al., 2023). Both of these baselines share the same operational parameters as follows: $\eta_G = \eta_H = 0.001, K = 10, \text{pns} = 10$, threshold = 0.05 and camcutoff = 0.001.

**Hyper-parameters for our proposed method GLIDE.** Our proposal has the following hyper-parameters: (1) The number of prior distributions $m$ used for the invariance test; and (2) $\gamma$ factor which controls the ratio of the data generated from weak interventions. Both of these parameters are concerned with the data sampling procedure. Other than these two parameters, there are two more thresholds which control the level of tolerable variance we consider as invariant and the confidence interval for the Conditional Independence Tests (CITs). These thresholds are set at $10^{-3}$ and 0.05, respectively, across all experiments reported in this study. We recommend tuning the former threshold depends on the nature of the data whereas the latter should remain unchanged.

Table 2: **Performance on real-world datasets.** N/A denotes no results within 48 hours.

| Metrics | Method | Sachs (11) | Insurance (27) | Water (32) | Alarm (37) | Barley (48) | Pathfinder (186) | Munin (1041) |
|---|---|---|---|---|---|---|---|---|
| **SHD** | GIES | $14.0 \pm 0.0$ | $36.0 \pm 0.0$ | $49.0 \pm 0.0$ | $44.0 \pm 0.0$ | $65.0 \pm 0.0$ | $1156.0 \pm 0.0$ | $1235.0 \pm 0.0$ |
| | PC | $10.3 \pm 0.7$ | $33.7 \pm 0.7$ | $54.3 \pm 0.7$ | $35.7 \pm 2.4$ | $58.3 \pm 1.7$ | N/A | N/A |
| | **GLIDE** | $\mathbf{5.2 \pm 0.4}$ | $\mathbf{18.0 \pm 2.8}$ | $\mathbf{41.6 \pm 1.8}$ | $\mathbf{27.8 \pm 2.0}$ | $\mathbf{45.8 \pm 2.4}$ | $\mathbf{59.1 \pm 1.9}$ | $\mathbf{883.2 \pm 21.8}$ |
| **Spurious rate** (%) | GIES | $34.8 \pm 0.0$ | $31.9 \pm 0.0$ | $35.8 \pm 0.0$ | $44.3 \pm 0.0$ | $35.8 \pm 0.0$ | $87.0 \pm 0.0$ | $42.4 \pm 0.0$ |
| | PC | $4.2 \pm 4.1$ | $\mathbf{0.0 \pm 0.0}$ | $17.1 \pm 2.1$ | $27.7 \pm 3.6$ | $17.8 \pm 1.9$ | N/A | N/A |
| | **GLIDE** | $\mathbf{0.0 \pm 0.0}$ | $3.6 \pm 2.7$ | $23.0 \pm 0.9$ | $\mathbf{13.1 \pm 0.9}$ | $\mathbf{8.2 \pm 1.1}$ | $\mathbf{1.9 \pm 0.5}$ | $\mathbf{1.8 \pm 0.2}$ |
| **Runtime** (seconds) | GIES | $\mathbf{1.5 \pm 0.4}$ | $\mathbf{2.7 \pm 0.6}$ | $\mathbf{3.1 \pm 0.6}$ | $\mathbf{3.3 \pm 0.4}$ | $\mathbf{3.4 \pm 0.8}$ | $\mathbf{19.3 \pm 0.4}$ | $\mathbf{61.5 \pm 15.2}$ |
| | PC | $52.0 \pm 1.8$ | $450.3 \pm 19.4$ | $573.8 \pm 16.1$ | $503.2 \pm 21.2$ | $761.1 \pm 9.5$ | N/A | N/A |
| | **GLIDE** | $23.0 \pm 1.1$ | $34.2 \pm 0.9$ | $49.1 \pm 3.3$ | $43.3 \pm 0.4$ | $61.7 \pm 2.0$ | $197.0 \pm 21.2$ | $6200.3 \pm 88.7$ |

## C.2 Main result: Synthetic categorical data

Categorical data are generated in the same manner as continuous data in Section 5.1. The only difference is that instead of using Gaussian models, we first randomize conditional probabilities for each node and then use Gibbs sampling to simulate the data. The number of categories are randomly chosen from 2 to 5 for each variable. We compare our proposal with two competitive baselines, GIES and PC, while excluding the rest since they fail to produce meaningful results for comparison (i.e., very poor performance). To focus on settings where the PC baseline can produce results within the time limit of 48 hours, we restrict our evaluation to two classes of graphs where the number of edges is (i) 3×, and (ii) 4× the number of nodes, which ranges between 100 and 500.

The results are reported in Figure 6 which shows that **GLIDE** consistently outperforms the baselines in terms of SHD with substantial gaps of 10.49% and 27.95% in the ($\mathbf{E} = 3\mathbf{V}$) case; and 11.56% and 13.3% in the ($\mathbf{E} = 4\mathbf{V}$) case. In both cases, **GLIDE** is the second best in terms of spurious rate, increasing the false causal detection (spurious) rate of the best baseline (PC) by a mere margin of 4%. In exchange, **GLIDE** achieves a 30× faster processing time than PC. In contrast, GIES achieves the fastest running time but incurs 40% spurious rate (i.e., low reliability). Overall, **GLIDE** has the best trade-off between scalability (processing time) and performance (SHD, spurious rate).

## C.3 Main result: Real-world graphs

Table 2 reported the performance of PC, GIES, and **GLIDE** on 7 real-world graphs provided by the Bnlearn open library Scutari (2009). Note that GIES is a deterministic method and has zero deviation across different seeds on the same test case. It is observed that **GLIDE** achieves the best SHD performance across 7/7 datasets and also achieves the best spurious rate in 5/7 datasets, especially on large datasets such as Barley, Pathfinder, and Munin. On the largest dataset (Munin), **GLIDE** achieves a spurious rate of 1.8% which is remarkably better than GIES' (42.36%). Again, **GLIDE** has the best balance between performance and scalability.

## C.4 Ablation studies

### C.4.1 Studies on the influence of $m$ and $\gamma$

The number of prior distributions $m$ corresponds to the number of environments/sub-datasets that our proposal generates. The higher this figure, the better the performance of invariance test. However, this comes with a trade-off on time consumption: the invariance test loops through all generated sub-datasets; therefore, it consumes increasing time linearly with the increase of $K$. On the other hand, $\gamma$ factor virtually controls the error in estimations produced by sub-datasets because $\gamma$ bounds the volume of the downsampled datasets.

In this study, we conduct experiments on the 100-variable Erdos-Renyi categorical data graph and continuous data graph. In each data model, we examine different values of $\gamma \in \{0.2, 0.4, 0.6, 0.8\}$, along with increasing

Table 3: **Influence of hyper-parameters.** Mean value and confidence interval 95% are reported.

| $m$ | $\gamma$ | SHD | Spurious rate (%) | Runtime (minutes) |
|---|---|---|---|---|
| 10 | 0.2 | $122.2 \pm 9.788$ | $5.91 \pm 1.678$ | $1.95 \pm 0.037$ |
| | 0.4 | $127.4 \pm 10.31$ | $5.41 \pm 0.951$ | $2.02 \pm 0.127$ |
| | 0.6 | $113.6 \pm 2.525$ | $4.64 \pm 1.01$ | $2.39 \pm 0.113$ |
| | 0.8 | $121.0 \pm 7.717$ | $4.19 \pm 1.552$ | $2.45 \pm 0.114$ |
| 20 | 0.2 | $116.8 \pm 9.139$ | $5.53 \pm 0.841$ | $4.54 \pm 0.221$ |
| | 0.4 | $109.8 \pm 8.796$ | $4.30 \pm 0.792$ | $4.18 \pm 0.086$ |
| | 0.6 | $123.0 \pm 2.772$ | $5.96 \pm 0.957$ | $3.88 \pm 0.098$ |
| | 0.8 | $112.8 \pm 5.958$ | $3.27 \pm 1.448$ | $3.93 \pm 0.095$ |
| 30 | 0.2 | $117.0 \pm 4.679$ | $5.19 \pm 0.505$ | $5.67 \pm 0.068$ |
| | 0.4 | $112.8 \pm 7.874$ | $4.37 \pm 0.613$ | $5.66 \pm 0.012$ |
| | 0.6 | $102.0 \pm 6.263$ | $3.85 \pm 1.603$ | $5.58 \pm 0.218$ |
| | 0.8 | $106.0 \pm 9.461$ | $4.15 \pm 0.981$ | $5.75 \pm 0.194$ |
| 40 | 0.2 | $114.0 \pm 5.714$ | $4.79 \pm 0.753$ | $7.44 \pm 0.032$ |
| | 0.4 | $112.8 \pm 5.088$ | $3.94 \pm 0.529$ | $7.50 \pm 0.028$ |
| | 0.6 | $108.6 \pm 8.072$ | $3.97 \pm 1.536$ | $7.30 \pm 0.051$ |
| | 0.8 | $108.8 \pm 4.444$ | $4.03 \pm 1.279$ | $7.36 \pm 0.048$ |
| 50 | 0.2 | $100.8 \pm 2.432$ | $5.03 \pm 0.518$ | $8.97 \pm 0.024$ |
| | 0.4 | $106.6 \pm 5.834$ | $4.64 \pm 0.958$ | $9.35 \pm 0.061$ |
| | 0.6 | $99.01 \pm 6.261$ | $3.68 \pm 1.589$ | $9.21 \pm 0.021$ |
| | 0.8 | $107.6 \pm 7.606$ | $4.61 \pm 0.955$ | $9.22 \pm 0.039$ |

values of $m \in \{10, 20, 30, 40, 50\}$. The input data volume is 10000 and we examine 10 runs to report mean values and 95% confidence intervals as in Table3. The results show a significant decrease (averagely, over 18%) in SHD when we increase the number of environments $m$ from 10 to 50. However, such performance comes at the expense of Runtime. As we discussed, the time complexity is linear with $m$, thus, we see that the runtime is about 5 times higher when $m$ is 5 times higher. As for $\gamma$. The increase of $\gamma$ does not guarantee better performance, as can be seen in settings $m = 10$, $m = 30$, and $m = 50$ where the best $\gamma$ is neither the smallest nor the biggest. Despite the effect of $\gamma$ is secondary to that of $m$, a good $\gamma$ can boost the performance roughly $6 - 8\%$ in term of SHD without incurring additional time complexity.

### C.4.2 Studies on different topologies

In this section, we test the performance of baselines and our proposal on Scale-free graphs and Bipartite graphs (Zheng et al., 2018). The code used to generate the datasets for experiments is introduced by Zheng et al. (2018). We also evaluate our proposal and baselines in normal and extreme cases - the former has the number nodes increasing from 100 to 500 and the number of edges equals the number of nodes, the latter has the number of nodes fixed at 500 while the number of edges increases from 600 to 1000. In each case, we also generate linear Gaussian data and non-linear non-Gaussian data scenarios. The results are depicted in Figures 7 and 8 for Bipartite graphs and Scale-free graphs, respectively.

**Bipartite graphs.** Regarding Bipartite graphs, our proposal shows superior performance in almost all cases and data generation scenarios. As can be seen in Figure 7(a)-upper, our proposal achieves relatively similar to the strongest baseline (MLP-Notears) in terms of SHD and spurious rate, but costs over an order of magnitude smaller in runtime, about 15.68 times.

The performance gap is noticeable when it comes to the non-linear non-Gaussian data scenario – Figure 7(a)-lower. Our proposal now has a clear advantage in terms of SHD (roughly 10% performance gap) and spurious rate (17.21% performance gap) against MLP-Notears, while the runtime difference is as significant as the previous scenario. It is worth noticing that, none of the baselines can achieve both fast runtime and relatively

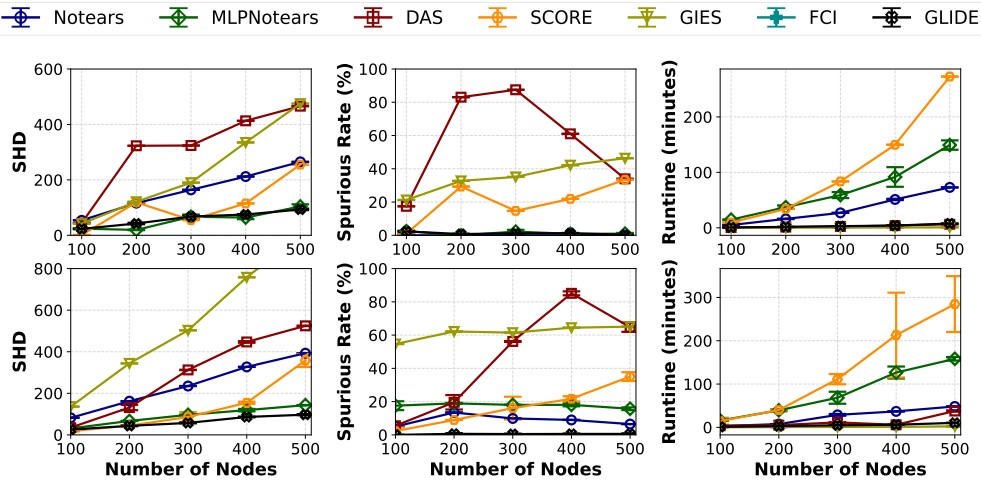

(a) **Evaluation on linear Gaussian (upper) and non-linear non-Gaussian (lower) data models.** The number of edges equals the number of nodes.

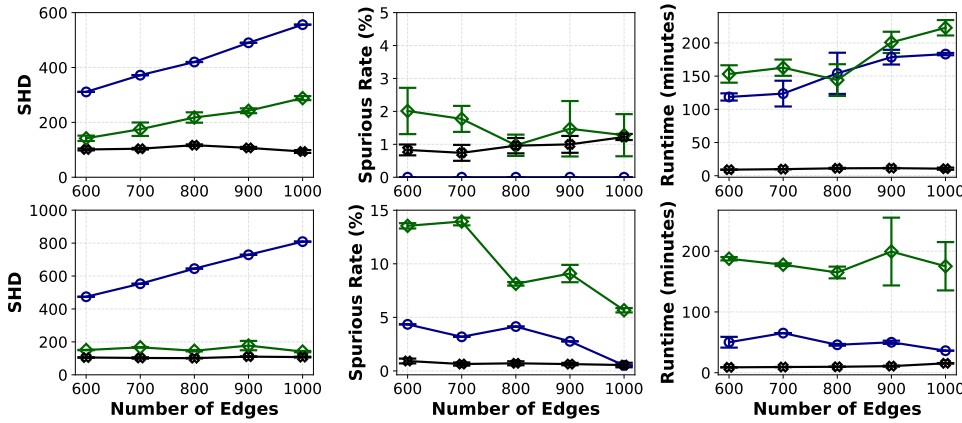

(b) **Evaluation on linear Gaussian (upper) and non-linear non-Gaussian (lower) data models at extremes.** The number of nodes is fixed at 500. Best baselines are selected to compare with our proposal.

Figure 7: **Performance on continuous data models in normal and extreme cases on Bipartite graphs.** Apply for all metrics: Lower is better.

high performance. For example, GIES and DAS have the same runtime as our proposal, but their performance in terms of SHD and spurious rate are significantly worse (at an aaverage of 61.55% and 46.22%, respectively).

As for the extreme cases in Bipartite graphs, we isolate best baselines, Notears and MLP-Notears, to compare with our proposal. The overall result in Figure 7(b) is that our proposal outperforms both baselines in both scenarios and in all metrics, albeit with an exception. Figure 7(b)-upper shows the result in linear Gaussian data scenario. We can see that our proposal has a noticeable gap to the other 2 baselines in both SHD (12.81% and 27.18% lower than MLP-Notears and Notears, respectively) and runtime (16.62 and 14.18 times less than MLP-Notears and Notears, respectively). However, Notears marginally outperforms our proposal in term of spurious rate, about 1%.

Figure 7(b)-lower shows that our proposal returns relatively stable performance even in the non-linear non-Gaussian data scenario with an average of 0.69% spurious rate. Furthermore, our proposal consistently outperforms both Notears and MLP-Notears in all cases in term of SHD. In contrast, Notears and MLP-Notears seem to be heavily affected by non-linearity in term of spurious rate. MLP-Notears has an average of 10.06% spurious rate – about 5 times higher than linear Gaussian data scenario. Notears also suffers an

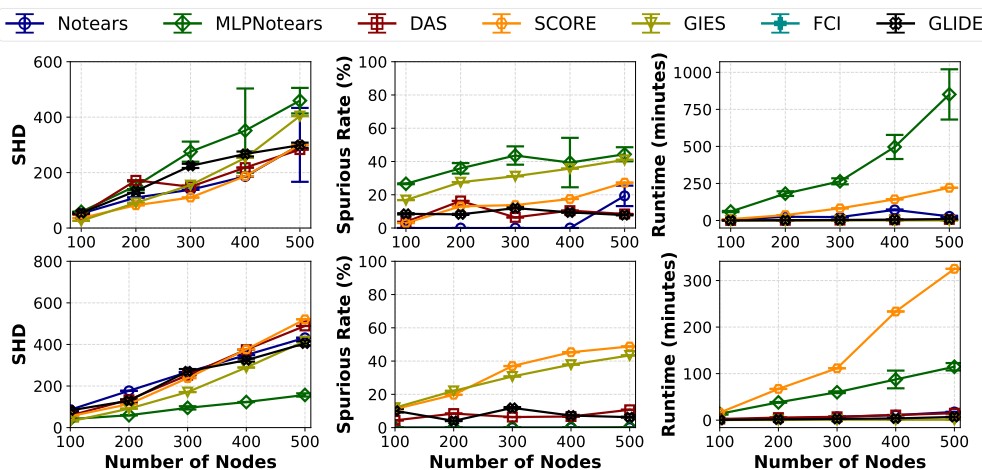

Figure 8: **Evaluation on linear Gaussian (upper) and non-linear non-Gaussian (lower) data models on Scale-free graphs.** The number of edges equals the number of nodes.

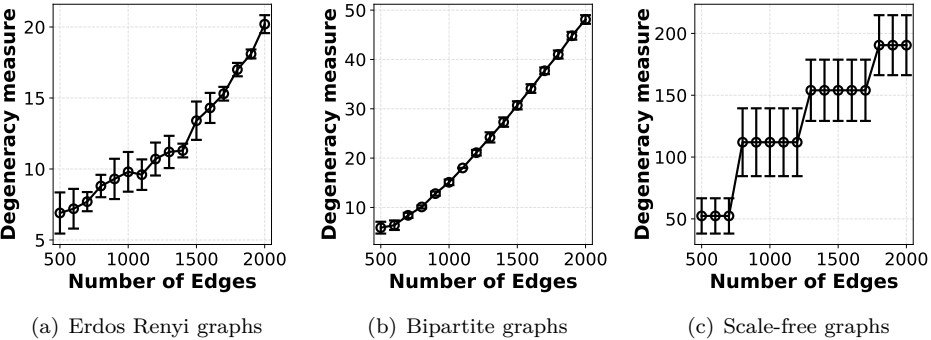

| (a) Erdos Renyi graphs | (b) Bipartite graphs | (c) Scale-free graphs |

Figure 9: **Degeneracy measure on different topologies.** The number of nodes is fixed at 500.

average 3% higher spurious rate. Interestingly, both baselines seem to benefit from the increasing number of edges in the graph as the results show an downward trend in spurious rate. However, it is worth noticing that Notears incurs a linearly increasing in SHD as the number of edges increases.

**Scale-free graphs.** Regarding Scale-free graphs, Figure 8 shows the results of algorithms in normal cases. Overall, despite not being the best performer in all metrics, **GLIDE** maintains a good trade-off between all metrics. Regarding the **L-G** case (Figure 8-upper) shows a different trends in performance of baselines compared to **GLIDE** in term of SHD. In details, the SHD of our proposal gradually increases until plateaus out in the 400 and 500-node graphs contrasting to other baselines (e.g., SCORE, and MLP-NOTEARS) that show an accelerating rate in SHD towards 500-node graphs. Furthermore, these baselines are noticeably worse than **GLIDE** when it comes to spurious rate (**GLIDE** produces an average of 5.52% less spurious relationships than SCORE, 28.57% than MLP-NOTEARS while being comparable to NOTEARS and DAS). As for the **nL-nG** case (Figure 8-lower), we have the same observation where our proposal consistently being comparable with other baselines in term of SHD, having a low spurious rate and runtime, simultaneously.

As regarding the extreme cases on the Scale-free graphs, when the number of edges exceeds 700, we encounter the following problem: the average number of parents for each node becomes exponentially large due to the nature of the graph. Such that the performance of MLP-Notears is heavily impacted: MLP-Notears cannot produce meaningful results within 48 hours. Our proposal - **GLIDE** is also impaired by this setting. The reason is that the data set is not sufficiently large to perform invariance test with adequate accuracy due to the exponentially large number of parents. Therefore, we do not include the report on performance of **GLIDE** as well as other baselines on this particular setting.

Table 4: **Degeneracy measures on real-world causal graphs.**

| Datasets | Sachs | Insurance | Water | Alarm | Barley | Pathfinder | Munin |
|---|---|---|---|---|---|---|---|
| Number of Nodes | 11 | 27 | 32 | 37 | 48 | 186 | 1041 |
| Degeneracy $p$ | 3 | 4 | 6 | 4 | 5 | 5 | 4 |

### C.4.3 Studies on the Degeneracy measure

In this ablation study, we investigate the degeneracy measure $p$ on different topologies and connect them to the performance of **GLIDE**. As mentioned in Section 4.3, the time complexity that **GLIDE** requires to find the plausible parent sets of all $d$ variables is $O(pd \cdot 3^{p/3})$. As such, we want to assert that, the degeneracy $p$ of most graphs is indeed insignificant compared to the number of variables $d$ of those graphs. To show this, we design the following ablation study: we generate DAG with a fixed number of nodes (500) and an increasing number of edges (500 to 2000). For each setting of the number of edges, we randomly generate 10 graphs and record their degeneracy measure. Notice that most baselines presented in this research are incapable of running on graphs with 500 nodes and more than 1000 edges – as we presented in Main text's Section 5 and Appendix C.4. However, for the sake of the argument of this section, we increases the number of edges to 2000 to investigate the range value of the degeneracy measure $p$. Figure 9 shows the mean and confidence interval 95% of the degeneracy measure on different topologies (Erdos-Renyi, Bipartite, and Scale-free graphs) as the number of edges increases.

Clearly, Bipartite graphs return the most stable degeneracy measure out of the three topologies. With very small variations, the degeneracy measures on increasingly dense Bipartite graphs demonstrate an almost linear growth. We speculate that this stable behavior of the degeneracy measure of the Bipartite graph benefits the performance of **GLIDE**, as we can see in Figure 7(b) where **GLIDE** consistently and noticeably outperforms the other two prominent baselines, especially in term of runtime. It is worth noting that at the (500-node, 1000-edge) setting, the degeneracy measure is roughly 15, which is much less than the number of nodes. This shows that, at cases where most baselines take hours to solve, **GLIDE** can still achieve almost quadratic time complexity (Section 4.3) and thus only takes minutes to recover the causal graph.

Regarding the Erdos-Renyi graphs, we can see it random nature manifest in a noticeable – yet not too significant – variation of the degeneracy measure. Nonetheless, these values (even at maximum, e.g., $p = 13$ at 1000-edge or $p = 22$ at 2000-edge graphs) are insignificant compared to the number of nodes. This partially explains the scalability of **GLIDE** on the Erdos-Renyi graphs as we reported in Main text's Section 5. Contrasting to the previous two, the degeneracy measure on Scale-free graphs shows an unstable behavior and a wide range of fluctuation. The degeneracy measure on 500-edge graphs upto 700-edge graphs has an average of $52.4 \pm 14.27$, which is highly unstable and generally much higher than that of the same setting but on other topologies. As the number of edges gradually reaches 700, the degeneracy has a sudden leap upto an average of $112 \pm 27.43$. This unusual (and perhaps, non-linear) behavior may stem from the implementation (Zheng et al., 2018) or may need further studies to address. Regardless, we can see that the range for degeneracy in Scale-free graphs are significantly higher than that in Erdos-Renyi or Bipartite graphs, which potentially affect the runtime of Algorithm 3. However, as we will see in Appendix C.4.4, the number of plausible sets in Scale-free graphs is still negligible compared to the number of nodes.

Finally, we also investigate the degeneracy of real-world causal graphs. These graphs are the same as the one presented in the Main text's Section C.3. As it turns out, real-world causal graphs are indeed sparse (see Table 4). In details, the degeneracy of the causal graph of these dataset are: (Sachs: 3, Insurance: 4, Water: 6, Alarm: 4, Barley: 5, Pathfinder: 5, Munin: 4). In which the Munin dataset has over 1000 variables. This shows that, in practice, we indeed often encounter large causal graphs that are sparse.

### C.4.4 Studies on the number of plausible parent sets

The number of plausible parent sets plays an essential role in the time complexity of our proposal. As per Algorithm 3 – an enhanced Bron-Kerbosch algorithm (Bron & Kerbosch, 1973), each plausible set corresponds to a maximal clique. Theoretical results from Bron & Kerbosch (1973) give us the bound for the number

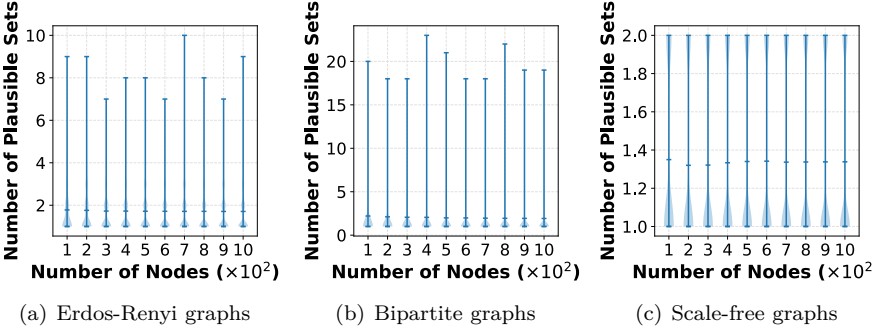

(a) Erdos-Renyi graphs      (b) Bipartite graphs      (c) Scale-free graphs

Figure 10: **Investigation on the number of plausible parent sets in different topologies.** The number of edges equals the number of nodes.

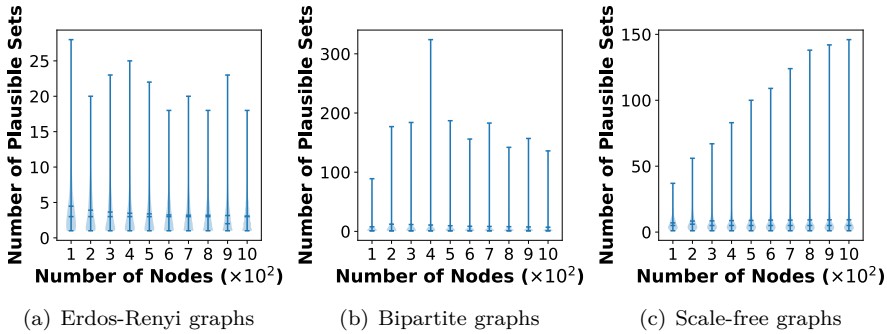

(a) Erdos-Renyi graphs      (b) Bipartite graphs      (c) Scale-free graphs

Figure 11: **Investigation on the number of plausible parent sets in different topologies.** The number of edges doubles the number of nodes.

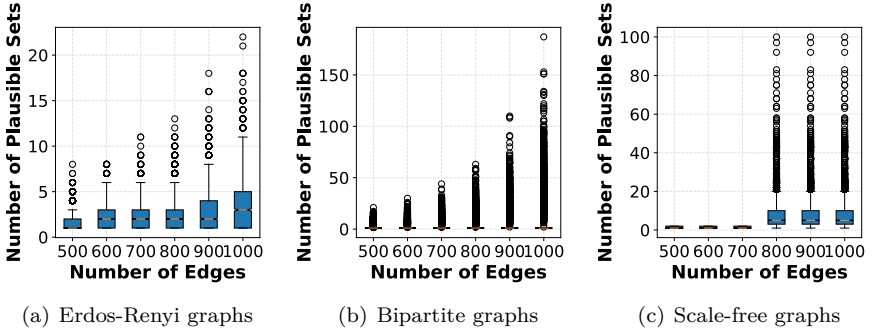

(a) Erdos-Renyi graphs      (b) Bipartite graphs      (c) Scale-free graphs

Figure 12: **Investigation on the number of plausible parent sets in different topologies.** The number of nodes is fixed at 500.

of maximal clique, which is $O((d-p) \cdot 3^{p/3})$ where $p$ is the degeneracy of the graph. As we have shown in Appendix Section C.4.3, $p$ is indeed insignificant compared to $d$ in the common Erdos-Renyi and Bipartite class of graphs. In this section, we empirically investigate the number of plausible parent sets in different graph scenarios. For each scenarios, we randomly generate 10 graphs using the source code provided by (Zheng et al., 2018) and perform Algorithm 3, then record the number of plausible parent sets of each node. We use the violin plot to show the distribution of the number of plausible sets at each setting. The maximum, minimum, mean, and median are reported.

Figure 10 shows the number of plausible sets on graphs whose number of nodes equals that of edges, which gradually increases from 100 to 1000. Notice that this setting mimics that of experiments that we reported in normal cases in Main text's Section 5.1 but the number of nodes grows from 100 to 500. As we can see

Table 5: **A case study on sortability.** Comparison with Varsort and R2sort.

| | **\|E\|** | **Varsort-ability** | **R2sort-ability** | **SHD** | | | **Spurious rate (%)** | | | **Time (min)** | | |
|---|---|---|---|---|---|---|---|---|---|---|---|---|
| | | | | **Varsort** | **R2sort** | **Ours** | **Varsort** | **R2sort** | **Ours** | **Varsort** | **R2sort** | **Ours** |
| L-G | 500 | 0.98 | 0.76 | **92** | 814 | 249.12 | 12.74 | 55.28 | **1.98** | 11.01 | **8.56** | 27.25 |
| L-G | 1000 | 0.98 | 0.92 | 739 | 4600 | **539.6** | 40.83 | 81.43 | **5.78** | 10.76 | **10.21** | 15.19 |
| nL-nG | 500 | 0.88 | 0.32 | 587 | 1316 | **340.66** | 50.99 | 68.02 | **4.18** | 27.57 | 26.21 | **7.25** |
| nL-nG | 1000 | 0.87 | 0.23 | 1642 | 4157 | **710.66** | 59.35 | 78.41 | **7.82** | 22.28 | 21.99 | **5.43** |

in Figure 10(a), the number of plausible sets on Erdos-Renyi graphs peaks at 10, with mean being roughly 2. This supports the excellent runtime of **GLIDE** as we reported. When it comes to Bipartite graphs, the number of plausible sets is approximately double that in the same settings on the Erdos-Renyi graphs. However, in all cases, this figure is much lower than the number nodes. Noticably, in the case of Scale-free graph, we see an almost similar distribution of the number of plausible sets across graphs with different nodes. In details, most nodes have only 1 plausible parent set, and other have 2. This observation combined with nature of the Scale-free graphs, suggest that each plausible set contains a large number of nodes. Consequently, given that the observational data volume is limited at 10000, the accuracy of the invariance test of **GLIDE** reduces significantly, which explains the performance that we reported in Appendix Section C.4.

We also conduct experiments on graphs that are twice as dense (the number of edges is doubled that of nodes) as in the previous setting, and the results are reported in Figure 11. Interestingly, while the number of plausible sets on Erdos-Renyi roughly doubles (both maximum and medium) compared to the previous setting, the that on Bipartite graphs increases significantly. In details, in most setting, the maximum number of plausible ranges from under 100 (on 100-node graphs) to over 300 (on 400-node graphs), which is innegligible compared to the number of nodes. Nontheless, these figures are still strictly less than the number of nodes. It is worth noting that most nodes in the graphs have about 10 to 15 plausible parent sets. As for the Scale-free graphs (Figure 11(c)), despite the median number of plausible sets is 10, the maximum increases linearly with the number of nodes and then plateaus out at about 150 when the number of nodes reaches 900.

Lastly, we study the number of plausible sets at extreme scenarios (as in Main text's Section 5.1): we fix the number of nodes at 500, and increase the number edges from 500 to 1000. The results are depicted by box plot in Figure 12. The number of plausible sets on the Erdos-Renyi graphs is strictly less than 25 even in the hardest setting (1000-edge graphs) while the mean value is roughly 3.8 and 75%-percentile is at 5. In contrast, the number of plausible sets on the Bipartite graphs grows non-linearly, reaching a maximum (outlier) 180 on 1000-edge graphs. However, since most nodes do not have more than 5 plausible parent sets, we see that the outliers have little effect on the overall performance of **GLIDE**, as can be seen in Figure 7(b). On the other hand, the number of plausible sets on Scale-free graphs is very unstable, and (interestingly) has the same trend as the degeneracy (reported in Figure 9(c)). In details, when the number of edges is below 700, we have the number of plausible sets ranges from 1 to 3 with the mean 1.4. But when the number of edges exceeds 700, despite the mean of 5 and 75%-percentile of 10, the maximum number of plausible sets may reach to over 100.

### C.4.5  Independence of varsortability

In this section, we include baselines Varsort (Reisach et al., 2021) and R2sort (Reisach et al., 2023) into experiments, to show that the performance of **GLIDE** is independent of sortability measures as proposed in these works. Sortability is the score defined in the interval $[0, 1]$ where 0 implies extreme difficulty to find the topological order over the variables, and 1 implies extreme easiness. Following the same practice in the main text, we run Varsort and R2sort on Erdos-Renyi graphs, with 500 nodes and $\{500, 1000\}$ edges. The results are shown in Table 5. Generally, despite high var-sortability, Varsort is outperformed by **GLIDE** in 3/4 cases. Specifically, **GLIDE** is evidently superior when it comes to extremely large graphs, as can be seen in a stark reduction in term of both SHD (up to 2.2×) and Spurious rate (up to 8×). R2-sortability, on the other hand, plummets as we move from Linear Gaussian models to non-Linear non-Gaussian models, which results

Table 6: Varsort, R2sort, and GLIDE on real-world graphs.

| Datasets | Baselines | Sortability | SHD | Spurious rate (%) | Runtime (min) |
|---|---|---|---|---|---|
| Insurance | **Varsort** | 0.64 | 105 | 62.01 | **2.54** |
| | **R2sort** | 0.18 | 154 | 69.93 | 2.71 |
| | **GLIDE** | | **18** | **3.6** | 34.2 |
| Water | **Varsort** | 0.69 | 93 | 51.18 | 4.03 |
| | **R2sort** | 0.41 | 145 | 62.96 | **3.39** |
| | **GLIDE** | | **41.6** | **23** | 49.1 |
| Alarm | **Varsort** | 0.55 | 108 | 65.87 | **5.29** |
| | **R2sort** | 0.59 | 118 | 68.57 | 5.38 |
| | **GLIDE** | | **27.8** | **13.1** | 43.3 |
| Barley | **Varsort** | 0.76 | 127 | 54.33 | 7.78 |
| | **R2sort** | 0.36 | 181 | 61.05 | **7.5** |
| | **GLIDE** | | **45.8** | **8.2** | 61.7 |
| Pathfinder | **Varsort** | 0.79 | 1496 | 89.29 | **43.91** |
| | **R2sort** | 0.05 | 2433 | 92.94 | 44.94 |
| | **GLIDE** | | **59.1** | **1.9** | 197 |

in its poor performance. Overall, **GLIDE** is more preferable compared to both of these baselines in synthetic scenarios.

We also conduct the experiments on real-world scenarios (graphs provided in the bnlearn package), and the results are shown in Table 6. It is clear that real-world graphs are much harder to solve as evidently shown by the low sortability of both Varsort and R2sort. Regardless, **GLIDE** outputs a relatively good result with respect to such difficulties. Especially, in the Pathfinder dataset, **GLIDE** achieves a remarked low spurious rate of 1.9% whereas that of Varsort and R2sort are 89.29% and 92.94%, respectively. Furthermore, we also record a significantly lower in SHD, almost 42× lower than R2sort and 25× lower than Varsort.

In conclusion, the performance of **GLIDE** is relatively independent of sortability (Reisach et al., 2021; 2023) as empirical evidences have shown. Furthurmore, these experiments also show that our experimental design is extensive and not biased.

# D  SUPPORT MATERIALS

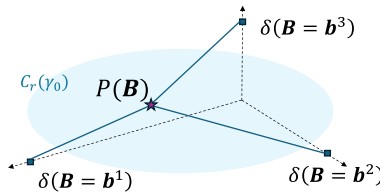 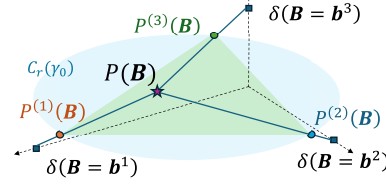 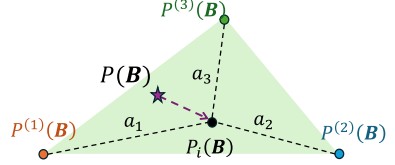

(a) Illustration of the space of $C_r(\gamma_0)$, original prior $P(\boldsymbol{B})$, and point mass distributions $\delta(\boldsymbol{B})$.

(b) Step 1: Computing the boundary points $P^{(i)}(\boldsymbol{B})$ via Eq.17, which forms a convex hull.

(c) Step 2: Performing weighted average of the boundary points to generate midpoints $P_i(\boldsymbol{B})$.

Figure 13: The procedure of generating prior distributions.

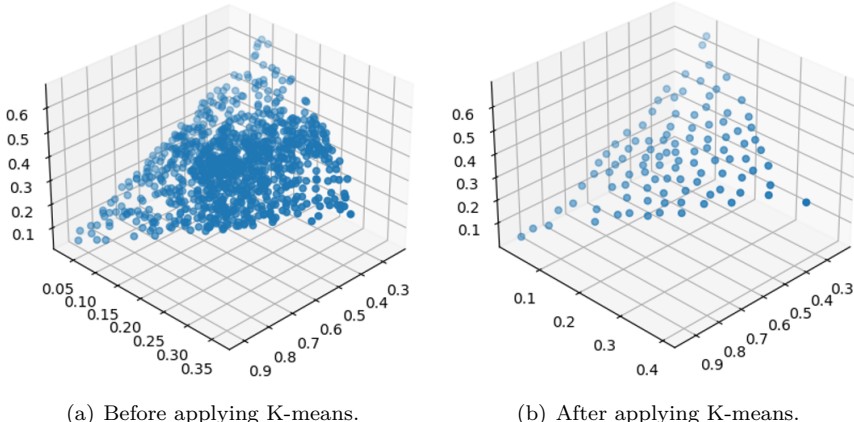

(a) Before applying K-means.        (b) After applying K-means.

Figure 14: **A mock-up experiment:** Representative distributions selection via K-means.

### D.1 On the generation of prior distributions

The fact that we can sample an infinite number of prior distributions $P_i(\boldsymbol{B})$ in $C_r(\gamma_0)$ is not helpful from the computational point of view. Because a larger number of prior distributions incurs a corresponding increase in the processing time of the invariance test. Furthermore, when the number of sampled prior distributions grows too large, it is inevitable that there exist similar prior distributions, which in fact does not improve the quality of the invariance test. On the other hand, an insufficient number of prior distributions reduces the reliability of the invariance test. Therefore, we wish to select only an adequate number of representative prior distributions within $C_r(\gamma_0)$.

Inspired by the aforementioned logic, we perform the following:

1. Sampling $P_i(\boldsymbol{B})$ where $i$ goes up to $10^4$ (Figure 14(a)) to overwhelmingly fill in the convex hull $C_r(\gamma_0)$. We use the Dirichlet distribution to sample weighted vectors $\boldsymbol{a}^{(i)}$.

2. Using the K-means algorithm with parameter $K = m$ to cluster them into $m$ partitions, each corresponds to a representative centroid. These centroids are the output prior distributions of the procedure (Figure 14(b)).

To this end, we have sampled $m$ prior distributions that act as variants of the original $P(\boldsymbol{B})$ and can be used to fuel the invariance test by applying Theorem 6.

### D.2 Further Discussion

**Parallelism Prospect.** Our proposed framework **GLIDE** is readily applicable to scenarios where the data is distributed across multiple, private local devices such as the popular federated learning setting. In such scenarios, **GLIDE** will benefit directly from the fact that the distributed datasets, which are presumed to be governed by the same causal model, are already admitting identical underlying conditional distributions. As such, **GLIDE** can use those local datasets as synthetic augmented datasets which are essential to the effect-cause distributional invariance test. This can save time and help avoid unnecessary sampling errors.

**Limitations.** The proposed framework **GLIDE** has the following limitations:

**1.** Our performance partially relies on how well Algorithm 2 can find the source nodes for the basis. As previously mentioned, it is not guaranteed that Algorithm 2 will find all the basis variables. However, our extensive experiments have shown that when the number of data augmentations increases, the chance that non-source nodes are included in the basis set is lessen.

**2. GLIDE**'s performance might be hampered on scenarios where the observational data is insufficient to discern the true complexity of the underlying causal graph – as can be seen with scale-free graphs.

## E REPRODUCIBILITY

**Software.** Our implementation is in Python. The requirements include the installation of the causal-learn package (Zheng et al., 2024) for the use of CITs, scikit-learn Python package, along with pandas and numpy, which are standard libraries commonly used in Python.

> **Support Module for Continuous Data.** For the use of our proposal for continuous data models, we use a Discretizer module provided by the Python standard scikit-learn library. The number of discretizing bins is fixed at 4 and the width of bins is equal to capture the marginal distribution of each variable in the observational data. As for the categorical data models, this module is deactivated.

**Hardware.** All experiments are run on a machine with a 64-core Intel(R) Xeon(R) Gold 6242 CPU @ 2.80GHz. The running of GPU-based baselines is conducted on an NVIDIA GeForce RTX 4090.

