# OpenReview forum: "Causal Graph Learning via Distributional Invariance of Cause-Effect Relationship"
_TMLR — Accepted by TMLR_

### Review · Reviewer_gpEM · 2025-09-20

**Summary Of Contributions:**

**Summary:**

This work studies the problem of recovering a causal DAG from observational data. Relying on observational data alone limits one to only being able to recover the DAG up to Markov equivalence class, but it does allow one to probe causal relationships even in the absence of interventional experiments.

Existing approaches to this problem typically fall into one of three categories: constraint-based, score-based, and model-based approaches. All have drawbacks: constraint-based approaches can have exponential computational cost, score-based approaches rely on heuristic objectives, and model-based approaches can be unstable when the model assumptions are violated. The authors propose a new model-based method, which they claim mitigates all three above drawbacks.

The authors’ core insight is that if one assumes effect-cause conditional distributions, e.g., P(effect | cause), are invariant to changes in the prior causal distribution P(cause), then one can efficiently leverage this relationship. For example, under this assumption, if one can construct sub-sampled datasets that reflect different causal priors P(cause) then one can rule out cause-effect pairs by concluding that no causal relationship (cause -> X) exists for any variable X such that P(X | cause) is not invariant across the different data subsets.

Surprisingly, the authors argue that under the causal invariance assumption, not only can they rule out cause-effect pairs in the above manner, but actually that with high probability, one can provably recover the full set of cause-effect pairs from such a procedure.

The procedure relies on an existing method for recovering the Markov blanket of every variable from observational data alone. In order for the recovery guarantees for this pre-existing method to apply, the authors must assume that no latent confounders impact the causal relationships reflected in the observed variables, which is a reasonable assumption within causal discovery literature.

The resultant algorithm recovers the underlying causal graph with computational complexity quadratic in the number of observed variables, which is very efficient by the standards of causal discovery.

The core computational speedup (and the bulk of the computational cost of the proposed procedure) comes from the execution of a quadratic-time algorithm, which finds the Markov blankets (subsets, one per variable, which contain all parent nodes, child nodes, and child nodes’ other parents).

The work includes empirical results showing that on the synthetic and real-world datasets considered, the proposed novel method provides both speedup and a reduction in false-positive causal relationships.

**Weaknesses:**

My primary concern with the manuscript is that I am confused about some of the core theoretical claims. Here are some of my most fundamental confusions.
- Identifying the parent set through invariance. I am confused about the claim made in Theorem 1, namely that the parents of a node can be identified as the largest set of variables for which “when conditioned on, the probability of observing X becomes unchanged.” Isn’t it possible that X would be invariant when conditioned on other ancestors, not necessarily only parents? For example, consider some path graph A-> B -> X such that B is a copy of A. Then conditioning on A should have the same impact as conditioning on B. Are the Markov blankets somehow being leveraged to prevent considering non-parent ancestors? Are they implicitly constraining the candidate set $\mathcal{Z}$?
- Identifying parent sets despite only targeting recovery up to Markov equivalence. As acknowledged by the authors, “Finding the true causal graph is not possible without running intervention[s] since different graphs can entail the same set of statistical constraints induced by observational data” (footnote, page 1). Thus when using only observational data, one’s goal must be restricted to identifying causal DAGs up to Markov equivalence. How then are the parent sets being recovered in Theorem 1? Is there some notion of parent set up-to-equivalence that the authors are employing?
- Reliance on correctly identifying source nodes. Does the proof of Theorem 4 rely on the fact that B is assumed to be a source variable? The proof references the fact that B is a source variable, but as acknowledged by the authors, “finding the true sources is not possible without intervention.”

**Audience:**

Yes

**Audience Explanation:**

If the paper’s theoretical claims are correct, then I think that readers would be excited by a novel approach to causal discovery.

**Broader Impact Concerns:**

I am not concerned about the ethical implications of the proposed method.

**Claims And Evidence:**

No

**Claims Explanation:**

As discussed in the Weaknesses section of “Summary of Contributions,” I am confused about the correctness of some of the core theoretical claims.

If the authors can correct my confusions about the theoretical claims, I am open to changing this answer.

The empirical results provide evidence for the practicality of the method.

**Requested Changes:**

Some concrete changes that would improve readability would include:
- Clarifying that causal invariance is necessarily an assumption. When first introduced in the text, the authors’ presentation may mislead readers into thinking that causal invariance is a fact rather than an assumption. See for example, in the abstract “leveraging the fact that the distribution of an effect, conditioned on its causes, remains invariant to changes in the prior distribution…”, and the text on page 2, which states “Our approach leverages the invariance of the effect-cause conditional distribution … to changes in the prior cause distribution.”
- Minor: re-organize proofs so that each result only relies on previously-proven results. E.g. the conclusion of the proof of Theorem 4 seems to rely on Theorem 5.

Typos:
- In Theorem 1, the line beginning “For each variable…” does not seem to parse grammatically to me. Should the authors remove the word “then” following the comma, or is there some quantifier or modifier missing from the statement?

---

> ### Comment · Editors_In_Chief · 2025-09-24
> **Response to Reviewer gpEM**
>
> ### We thank the reviewer for providing detailed feedback, along with many insightful comments and questions.
>
> ### Regarding the weaknesses, we address each concern as follows:
> 1. **Identifying the parent set through invariance.** The Markov blanket of a node $X$ consists of its parents, children, and spouses. If non-parent ancestors exist in a Markov blanket of $X$, they must also be spouses of $X$ and are treated as such. In theorem 1, we show that with high probability we can detect and remove all spouses from a given plausible parent set $\boldsymbol{Z}$ by using the $\mathrm{Pruned}_{X}(\boldsymbol{Z})$ operator. In any case, we leverage the well-established GSMB algorithm to recover the Markov blanket of each node. Improving this algorithm will help improve our work directly but it will be orthogonal to our main focus here.
>
> 2. **Recovery up to Markov Equivalence.** With high probability, our method recovers the true parent sets with respect to one of the graphs within the Markov Equivalence class. In detail, Theorem 1 shows that with high probability, non-causal relationships are removed from plausible parent sets, which means that our output graph shares the same skeleton structure with the Markov Equivalence graph. Secondly, all V-structures in the graph is guaranteed to be descendants of the basis nodes (see Lemma 2) and therefore are correctly oriented by Theorem 1. In conclusion, with high probability, our method is guaranteed to find a graph within the Markov Equivalence class.
>
> 3. **Reliance on correctly identifying source nodes.** No, the proof of theorem 4, which concerns the downsampling ratio, need not assume that the basis variables are sources in the graph. We will revise the text to highlight this clarification. However, the overall performance of our method relies on how well the basis $\boldsymbol{B}$ covers the true source variables, which we mentioned in *Appendix D.2. Limitations*. Empirically, we estimate the sources by a maximum-sized basis set, which already results in higher performance compared to state-of-the-arts methods. This opens up space for even better performance if we improve the source-finding algorithm or integrate auxiliary information.
>
> ### Regarding requested changes:
> 1. **Clarifying that causal invariance is necessarily an assumption.** We agree and will address this in the revised manuscript.
> 2. **Minor: rearrangement of the proofs.** Actually, the very last sentence of the proof of Theorem 4 serves as a "conjunction" towards the proof of Theorem 5 only. We will remove this sentence, which would make the proofs of theorem 4 and 5 become independent.

---

> ### Comment · Editors_In_Chief · 2025-10-12
> **Manuscript update**
>
> Dear Reviewer gpEM,
>
> We'd like to update you that our manuscript has been revised and uploaded following yours and Reviewer pMr2's suggestions.
>
> There is one additional point that we would like to provide further clarification. Upon revision, we made it clear that the original statement under Assumption 1 is a direct consequence of the local Markov conditions. The wording "Assumption 1" is an editorial typo. We have corrected it to Observation 1 with additional elaboration in Remark 1. We also note that this editorial changes for better clarity does not affect the logical flow of our proposed work and its contribution in anyway.
>
> May we also know if our earlier response has addressed your concerns? Please let us know. We really appreciate your detailed review of our work.
>
> Best regards,
>
> Authors

---

### Review · Reviewer_pMr2 · 2025-09-24

**Summary Of Contributions:**

This work introduces a framework for causal graph discovery that exploits the invariance of the conditional distribution of effects given their causes to changes in the prior distribution of the causes, as implied by the local Markov property. The method employs a sampling strategy to generate data with altered cause distributions relative to the original sample, followed by an invariance test that evaluates whether the conditional distribution of effects given causes remains unchanged across these generated samples. Experimental results demonstrate that the proposed framework achieves performance that is superior or comparable to existing benchmarks, while maintaining quadratic time complexity.

**Additional Comments:**

None

**Audience:**

Yes

**Audience Explanation:**

The experimental results show favorable performance of the proposed framework in uncovering causal graphs from observational data. Although it does not outperform MLP-NOTEARS in some settings, this is a reasonable trade-off given the significant gains in computational efficiency.

**Claims And Evidence:**

Yes

**Claims Explanation:**

I found this paper engaging and nice to read. The main idea of testing invariance across generated samples seems a clever and scalable approach to causal discovery. The theoretical developments are carefully made and presented and appears to have broader applications beyond causal discovery, such as the proposed sampling procedure. While several theoretical compromises are made for practical implementation, for instance, substituting basis variables for source variables and assuming sparsity in the graph, these choices are well justified, clearly explained, and applicable to a broad class of graphs.

**Requested Changes:**

My suggestions primarily concern the presentation. Some definitions and notations are either inconsistent or not clearly specified. In addition, the exposition does not always follow the order of implementation steps, which makes the paper somewhat less straightforward to read:

1. It is not entirely clear whether Assumption 1 is truly an assumption or rather a property implied by the local Markov property. It seems more like the latter. By contrast, the statement “Accordingly, our proposed method relies on the contrapositive: if invariance does not occur, causality does not exist” functions as an assumption, but it is not explicitly emphasized or formalized outside of the introduction.
2. The notation is inconsistent in several places, which makes the context difficult to follow. For instance, the paper appears to use $\boldsymbol{Z}_i \in \mathcal{Z}_0$ and $\boldsymbol{Z} \in \mathcal{Z}_0$ interchangeably, which can be confusing. In Equation (1), $\prune_X(\boldsymbol{Z}_i)$ is defined as the operator that removes all $Y \in \boldsymbol{Z}$ such that $\exists \boldsymbol{Z}' \in \boldsymbol{Z}_i : X \perp Y \mid \boldsymbol{Z}_i$. It is unclear what the $\boldsymbol{Z}$ in $Y \in \boldsymbol{Z}$ refers to, is this a typo, or does it denote a different set from $\boldsymbol{Z}_i$? Also, does $\boldsymbol{Z}'$ include the empty set? In addition, the distinction between Equation (1) and Equation (3) is not clear, as they appear to be repetitive. Moreover, Equation (1) uses $\boldsymbol{Z} \in \mathcal{Z}_0$, whereas Equation (3) uses $\boldsymbol{Z}_i \in \mathcal{Z}_0$, which adds to the confusion.
3. The meaning of $\mathbb{V}_{P_{+}(\boldsymbol{X})} \sim \mathcal{P}\left[P_{+}\left(X \mid \boldsymbol{Z}_i\right)\right]=0$ in Equation~(1) in not stated or explained.
4. What is the intuition behind initializing $\boldsymbol{B}$ with $X \in \boldsymbol{V}$ having the smallest $|\Phi(X)|$? It seems that the selected $X$ is the variable least connected with others in the graph, but it is not clear why this makes $X$ a favorable candidate for inclusion in $\boldsymbol{B}$.
5. Could you expand on this ``To guarantee this, $D_i$ must be a downsampled version of $D$ via sampling with no replacement to avoid introducing duplicates and hence, false causal biases into $D_i$.''? In particular, why is sampling without replacement necessary in this context?
6. What does $|D|/|D_i|$ mean? Does it denote the ratio of unique observations, or something else?
7. Below Equation~(3), what does this line mean? ``Furthermore, it is worth noting that $\pa[X]\in\Z_0$ is always guaranteed while $\boldsymbol{Z}_i\notin \pa[X]$ only exists in $\Z_0$ in some specific scenarios.''
8. Clarify what is meant by ``(empirical) sparsity'' in the statement ``Using this invariance test and exploiting an (empirical) sparsity of most causal graphs''? An expanded description or discussion would be helpful, particularly in relation to the application scenarios of the proposed framework.

Therefore, in general I request changes:
1. Providing an algorithm summarizing the proposed framework for causal discovery. Make clear the applicable scenario, and mark where theoretical compromises are made.
2. Use consistent notation throughout and to clearly define each notation when first introduced, so that the presentation is fully self-contained.
3. Expand on key definitions.

---

> ### Comment · Editors_In_Chief · 2025-10-06
> **Response to Reviewer pMr2**
>
> We thank the reviewer for your detailed feedback and many constructive comments. We address each of your concerns as follows:
> 1. Indeed, Assumption 1 follows directly from the local Markov condition. The intended meaning is that if changes to $P(\boldsymbol{X})$ arise solely from changes in the marginal distributions of its parent variables, then the induced conditional distribution $P(X | \mathrm{Pa}[X])$ remains unchanged as a consequence of the local Markov condition. We realized that this statement became less clear after prior edits for conciseness, and that labeling it as an "Assumption" may have added to the confusion. We have therefore revised the manuscript to restate it as an Observation for clarity.
>
> 2. We apologize for the cluttering notations. We want to clarify that $\mathcal{Z}$ is the set of plausible parent sets. An item $\boldsymbol{Z}$ in $\mathcal{Z}$ denotes a candidate parent set for $X$. In the scope of Theorem 1, we use the subscript to indicate different parent sets within $\mathcal{Z}$. Later, in the text description of the prune operator $\mathrm{Prune}_X(\boldsymbol{Z}_i)$, we accidentally omit $i$ which causes a typo that adds to the confusion. Also, in that description, $\boldsymbol{Z}'$ can be the empty set. Later, Eq. (3) provides a (perhaps redundant) mathematical expression for this text description of the pruning operator. We also accidentally omitted the subscription $i$ in this equation. These have all been fixed in our updated manuscript.
>
> 3. $\mathbb{V}_{P(\boldsymbol{X}) \sim \mathcal{P}}[P(X | \boldsymbol{Z}_i)]$ denotes the variance of the induced $P(X | \boldsymbol{Z}_i)$ over the random choice of the joint distribution $P(\boldsymbol{X})$. Such joint distribution $P(\boldsymbol{X})$ is obtained via making (random) changes only to the source variables in the original joint distribution $P(\boldsymbol{X})$ that underlines the observation data. Later, we also show that although $P(\boldsymbol{X})$ remains latent, we can still find its source variables and apply distributional changes to those via a downsampling scheme (see Theorems 4 & 5 in Section 4.2.2). We have added a clarification statement for this in the updated manuscript.
>
> 4. The intuition is that given a causal graph, the set of dependence nodes for a source does not include other sources while most non-source nodes are dependent on all their upstream (including all source nodes) and downstream nodes. Thus, the set of dependence nodes for a source node will not be larger than that of a non-source node. We formalize this intuition in Lemmas 1 and 2 as part of the proof for Theorem 3 in Appendix A3. We have added this intuition statement following the statement of Theorem 3 in the main text of our updated manuscript.
>
> 5. Sampling without replacement ensures that no new duplicates are added to the re-sampled dataset. Intuitively, because this operation only removes information rather than generating new samples, it cannot introduce any spurious causal patterns into the data. We mentioned this in the first three lines in Section 4.2.2.
>
> 6. $|D|/|D_i|$ denotes the downsampling rate. We mention this in the second last sentence of the paragraph preceding Theorem 4. It is measured as the ratio of the volume of the original observational data $D$ over the volume of the sampled data $D_i$.
>
> 7. This means by construction $\mathcal{Z}_0$ contains all variable sets $\boldsymbol{Z}$ for which $\mathbb{V}[P(X | \boldsymbol{Z})] = 0$ when random changes are made to the marginal distribution over source variables; and this includes the true causal parent of $X$ (due to the local Markov condition as stated in Observation 1). There might also be a few other spurious candidates in $\mathcal{Z}_0$ but usually, they are special sets that are related to $X$, e.g., subsets of $X$'s Markov blanket that contains $\mathrm{Pa}[X]$. Such special cases can be removed with our Prune operator. It appears this latter part has been accidentally omitted from the original statement. We have added it back in the updated manuscript.
>
> 8. Empirical sparsity refers to the observation that in most benchmark datasets used for causal discovery evaluation, the augmented bidirectional graph (see Theorem 7) indicating whether two nodes are in each other's Markov blanket is p-degenerate (as defined in Definition 4) with a small p. This property allows us to leverage the depth-first search algorithm by Bron and Kerbosch on the augmented graph to efficiently enumerate all plausible parent sets in O(d^2), given the Markov blanket of each node in the original graph. As the sentence that refers to this "empirical sparsity" is in the abstract, we do not have the context to expand thoroughly on this point. In the updated manuscript, we will add a footnote on the first page to elaborate on this.
>
> **Regarding the requested changes:** We will provide a summarized pseudo-code for GLIDE.

---

### Review · Reviewer_Gmtf · 2025-10-23

**Summary Of Contributions:**

This paper introduces GLIDE, a new framework for causal discovery from observational data1. Its core idea is that the conditional distribution $P(\text{effect} | \text{causes})$ is invariant to changes in the prior $P(\text{causes})$. GLIDE proposes a novel "invariance test" that uses strategic downsampling to simulate different priors and check if the variance of $P(X|Z)$ is near-zero. By combining this test with Markov blanket identification and graph sparsity assumptions, it efficiently finds parent sets, claiming an $O(d^2)$ complexity. Experiments show it matches or beats existing methods on accuracy while being up to 25x faster.

**Key Strengths:**

- Novel application of distributional invariance principles to causal discovery
- Interesting downsampling approach
- Comprehensive empirical evaluation across multiple graph types and datasets
- Clear computational complexity analysis with claimed significant improvements

**Key Weaknesses:**

- Gaps in theoretical proofs, particularly Theorem 1 which forms the foundation
- Limited principled guidance for hyperparameter selection (m, γ, ε parameters)
- Unclear positioning relative to other invariance-based causal discovery methods

**Audience:**

Yes

**Audience Explanation:**

Causal discovery is a fundamental problem in machine learning with broad applications across scientific domains. The paper tackles scalability issues that limit practical deployment of causal discovery methods, which is highly relevant to the TMLR audience. The distributional invariance perspective, while building on existing work, offers a novel algorithmic approach that could inspire further research. The claimed computational improvements, if validated, would be significant for practitioners working with large datasets. Even with the theoretical and experimental limitations, the core ideas are sufficiently interesting to warrant attention from researchers working on causal inference, graphical models, and related areas.

**Broader Impact Concerns:**

This work presents a methodological improvement to causal discovery algorithms, focusing on computational efficiency and scalability rather than introducing new application domains or fundamentally changing how causal discovery is used. The potential risks (misinterpretation of causal relationships, over-confidence in results) are inherent to all causal discovery methods and are not specifically amplified by this approach.

**Claims And Evidence:**

No

**Claims Explanation:**

The theoretical claims are not adequately supported due to gaps in the mathematical proofs:

**Theorem1:**
- Point 2 is insufficiently justified: The proof claims that if Z contains a child Y of X, then P(X|Z) has variance > 0 across different source priors. The reasoning provided that P(Y|Z\Y) "lacks X in its conditioning set"doesn't rigorously establish why this creates dependence on source variable distributions. The connection between d-separation properties and distributional variance across resampled datasets needs much stronger mathematical foundation.
- Point 4's spouse detection claim is unsubstantiated: The proof asserts that the PrunedX operator can reliably detect and remove spouse variables. The reasoning for why "X ⊥⊥ Y | Z'" would hold for spouses in the specific cases outlined is not convincingly established.
- Point 5 makes an unjustified leap: Why is Pa[X] necessarily the *unique* maximum-sized set among PrunedX(Z₀)? The proof doesn't rule out multiple sets of equal maximum size
  - The proof fails to rule out the following critical possibilities:
    - That another pruned, invariant set exists with the same size as Pa[X]
    - That a set Z containing Pa[X] and a spouse Y happens to pass the invariance test (𝕍 ≈ 0), and Y is precisely the "pruning failure" case shown in Figure 5. In this scenario, this set Z would be larger than Pa[X], causing the algorithm to fail.

**Theorem 2:**
- The induction only shows |B(G)| ≤ |S(G)|, but proving the maximum basis size equals the number of sources requires showing equality, not just inequality

**Theorem 3**
- Lemma 1's claim about nodes with minimum dependents being sources lacks rigorous justification
- The argument about Type-1 colliders and why there "must exist" a node Z with smaller |Φ(Z)| is not mathematically sound

**Practical Implementation:**
- The method shows substantial sensitivity to hyperparameters but provides limited guidance for practical parameter setting

**Requested Changes:**

**Critical Changes:**

1. **Fix theoretical foundations**: Rigorously prove Theorem 1, particularly Point 2's claim about variance behavior and Point 4's spouse detection mechanism.

2. **Provide principled hyperparameter selection**: Develop theoretical or empirical guidelines for setting m, γ, and ε parameters. The current ad-hoc approach limits practical applicability.

3. **Strengthen basis-finding algorithm**: Provide rigorous proof for Theorem 3, particularly Lemma 1's claim about minimum dependency nodes being sources.

**Changes that would strengthen the work:**

1. **Improve positioning relative to related work**: Better compare against other invariance-based causal discovery methods and explain the specific advantages of the proposed approach.

2. **Add failure case analysis**: Analyze scenarios where the method fails and provide guidance on detecting such cases in practice.

---

> ### Author Response · Authors · 2025-11-07
> **Response to Reviewer Gmtf (1/3)**
>
> We thank the reviewer for many insightful comments. We appreciate the reviewer’s comment regarding the clarity of the proof. The original version relied on an intuitive argument, which may have been too concise. We have now provided an alternative proof that is more detailed and rigorous.
>
> In our proof, we use $\mathbb{V}_{P_i(\boldsymbol{B})}[\cdot]$ and $\mathbb{V}[\cdot]$ equivalently to denote the variance of the input with respect to changes of the distribution of the source variables $P_i(\boldsymbol{B})$.
>
> ---
>
> First, we establish the following **auxiliary result** which will be leveraged to establish our main result.
>
> **Lemma**: $\forall \mathbf{R} \subseteq \mathbf{X}\setminus\mathbf{B}: P_i(\mathbf{R}\mid \mathbf{B}) = P(\mathbf{R}\mid \mathbf{B})$. Consequentially, $P_i(\mathbf{R}) = \int_{\mathbf{B}}P_i(\mathbf{R}\mid \mathbf{B})P_i(\mathbf{B})d\mathbf{B} = \int_{\mathbf{B}}P(\mathbf{R}\mid \mathbf{B})P_i(\mathbf{B})d\mathbf{B}$.
>
> **Proof**: We have $P(\mathbf{X}) = P(\mathbf{B})P(\mathbf{X}\setminus\mathbf{B}\mid\mathbf{B})$ where $P(\mathbf{X}\setminus\mathbf{B}\mid\mathbf{B}) = \prod_{X \in \mathbf{X}\setminus\mathbf{B}}P(X\mid\mathrm{Pa}(X))$ is constant. Therefore:
>
> $P_i(\mathbf{R}\mid \mathbf{B}) = \int_{\mathbf{X}\setminus(\mathbf{B}\cup\mathbf{R})}P_i(\mathbf{X}\setminus\mathbf{B}\mid \mathbf{B})d\mathbf{B} = \int_{\mathbf{X}\setminus(\mathbf{B}\cup\mathbf{R})}P(\mathbf{X}\setminus\mathbf{B}\mid \mathbf{B})d\mathbf{B} = P(\mathbf{R}\mid \mathbf{B}) \quad \blacksquare$
>
> ---
> We will now state and prove the main result.
>
> For each node X, suppose its Markov blanket has been identified (via [Edera et al. 2014]). We can use the pairwise dependence test (see $\mathrm{Pruned}_X(\cdot)$ operator in **Theorem 1**) to remove its spouses from its Markov blanket. Its candidate parent set is therefore a powerset of $\mathrm{Pa}(X) \cup \mathrm{Ch}(X)$.
>
> **Theorem**: Our key result then establishes that over the random choice $P_i(B)$ as prior over sources $B$: (1) the variance over the induced conditional $P_i(X \mid \mathrm{Pa(X)})$ is zero, $\mathbb{V}[P_i(X \mid \mathrm{Pa}(X))] = 0$; and (2) the variance over $P_i(X \mid \mathbf{Z})$ where $\mathbf{Z} \subseteq \mathrm{Pa}(X) \cup \mathrm{Ch}(X)$ and $\mathbf{Z} \neq \mathrm{Pa}(X)$ is not zero. This means we can use $\mathbb{V}[P_i(X \mid \mathbf{Z})]$ where $\mathbf{Z} \subseteq \mathrm{Pa}(X) \cup \mathrm{Ch}(X)$ to determine the true causal parents of $X$.
>
> **Proof**: Since we only perform our invariance test on non-source variables $X$, we consider the following tuple $(X,\mathbf{Z})$ where $\mathbf{Z} = \mathbf{Z}_1 \cup \mathbf{Z}_2$, $\mathbf{Z}_1 \cap \mathbf{Z}_2 =\emptyset$, $\mathbf{Z}_1 \cap \mathbf{B} = \emptyset$, and $\mathbf{Z}_2 \subseteq \mathbf{B}$.
>
> Then we have:
>
> $P_i(X\mid\mathbf{Z}) = P_i(X\mid \mathbf{Z}_1, \mathbf{Z}_2) =
> \frac{P_i(X, \mathbf{Z}_1, \mathbf{Z}_2)}{P_i(\mathbf{Z}_1, \mathbf{Z}_2)} \quad (*)$
>
> where the numerator is $\int_{\mathbf{B}\setminus\mathbf{Z}_2}P(X, \mathbf{Z}_1\mid \mathbf{B})P_i(\mathbf{B})d\mathbf{B}$
>
> and the denumerator is $\int_{\mathbf{B}\setminus\mathbf{Z}_2}P(\mathbf{Z}_1\mid \mathbf{B})P_i(\mathbf{B})d\mathbf{B}$
>
> These result follow from the aforementioned auxiliary result.
>
> This means if $\mathbb{V}[P_i(X\mid\mathbf{Z})]=0$, then $\frac{P_i(X,\mathbf{Z}_1,\mathbf{Z}_2)}{P_i(\mathbf{Z}_1,\mathbf{Z}_2)}$ must be a constant function $\alpha(X,\mathbf{Z}_1,\mathbf{Z}_2)$ with respect to $P_i(\mathbf{B})$. Given its expansion above, we must have:
>
> $P_i(X,\mathbf{Z}_1\mid\mathbf{B})=\alpha(X,\mathbf{Z})P_i(\mathbf{Z}_1\mid \mathbf{B})\Rightarrow P_i(X \mid\mathbf{Z}_1, \mathbf{B})=P_i(X\mid\mathbf{Z}_1,\mathbf{Z}_2,\mathbf{B}\setminus\mathbf{Z}_2)=\alpha(X,\mathbf{Z}_1,\mathbf{Z}_2)$,
>
> where $\alpha(X,\mathbf{Z}_1, \mathbf{Z}_2)$ is a scalar function independent of the choice $P_i(\mathbf{B})$ for source prior. Equivalently, this means knowing $\mathbf{B} \setminus \mathbf{Z}_2$ does not change the condition of $X$ on $\mathbf{Z} = (\mathbf{Z}_1, \mathbf{Z}_2)$. Following the definition of conditional independence, we have:
>
> $X \perp (\mathbf{B} \setminus \mathbf{Z}_2) \mid (\mathbf{Z}_1, \mathbf{Z}_2)$. Hence,
>
> $\mathbb{V}[P_i(X\mid \mathbf{Z})] = 0 \Rightarrow X \perp (\mathbf{B} \setminus \mathbf{Z}_2) \mid (\mathbf{Z}_1, \mathbf{Z}_2)$ (+)

---

> ### Author Response · Authors · 2025-11-07
> **Response to Reviewer Gmtf (2/3)**
>
> On the other hand, if $X \perp (\mathbf{B} \setminus \mathbf{Z}_2) \mid \mathbf{Z}_1, \mathbf{Z}_2)$ holds, $P(X \mid (\mathbf{B} \setminus \mathbf{Z}_2), \mathbf{Z}_1, \mathbf{Z}_2) = P(X \mid \mathbf{Z}_1, \mathbf{Z}_2)$. Plugging this into $(*)$, we have the numerator:
>
> $P_i(X, \mathbf{Z}_1, \mathbf{Z}_2)$
>
> $=\int_{\mathbf{B}\setminus\mathbf{Z}_2}P(X, \mathbf{Z}_1\mid \mathbf{B})P_i(\mathbf{B})d\mathbf{B}$
>
> $=\int_{\mathbf{B}\setminus\mathbf{Z}_2}P(X\mid\mathbf{Z}_1,\mathbf{Z}_2,\mathbf{B}\setminus\mathbf{Z}_2)P(\mathbf{Z}_1\mid \mathbf{B})P_i(\mathbf{B})d\mathbf{B}$
>
> $=\int_{\mathbf{B}\setminus\mathbf{Z}_2}P(X\mid\mathbf{Z}_1,\mathbf{Z}_2)P(\mathbf{Z}_1\mid \mathbf{B})P_i(\mathbf{B})d\mathbf{B}$
>
> $=\int_{\mathbf{B}\setminus\mathbf{Z}_2}P(\mathbf{Z}_1\mid \mathbf{B})P_i(\mathbf{B})d\mathbf{B}\times P(X\mid\mathbf{Z}_1,\mathbf{Z}_2)$
>
> $=P_i(\mathbf{Z}_1,\mathbf{Z}_2)\times P(X\mid\mathbf{Z}_1,\mathbf{Z}_2)$
>
> Thefore $P_i(X\mid\mathbf{Z})=P_i(X\mid\mathbf{Z}_1,\mathbf{Z}_2)=\frac{P_i(X, \mathbf{Z}_1, \mathbf{Z}_2)}{P_i(\mathbf{Z}_1, \mathbf{Z}_2)}=P(X\mid\mathbf{Z}_1, \mathbf{Z}_2)$
>
> which does not depend on $P_i(\mathbf{B})$. This means:
>
> $X \perp (\mathbf{B} \setminus \mathbf{Z}_2) \mid (\mathbf{Z}_1, \mathbf{Z}_2) \Rightarrow \mathbb{V}[P(X\mid \mathbf{Z})] = 0$ (++)
>
> Combining (+) and (++), we obtain a **bidirectional equivalence**:
>
> $\mathbb{V}_{P(\mathbf{B})}[P_i(X\mid \mathbf{Z}_1, \mathbf{Z}_2)] = 0 \Leftrightarrow X \perp (\mathbf{B} \setminus \mathbf{Z}_2) \mid \mathbf{Z}_1, \mathbf{Z}_2$ (+++)
>
> Assuming that $\mathrm{Pa}(X) \cap \mathrm{Spouse}(X) = \emptyset$ and $\mathrm{Spouse}(X) \cap \mathbf{B} = \emptyset$, the above **bidirectional equivalence** in (+++) has these direct consequences:
>
> (a) $\mathbf{Z}_1 \cup \mathbf{Z}_2 = \mathrm{Pa}(X) \Rightarrow X \perp (\mathbf{B} \setminus \mathbf{Z}_2) \mid (\mathbf{Z}_1 \cup \mathbf{Z}_2) \Rightarrow \mathbb{V}[P_i(X\mid \mathbf{Z})] = 0$
>
> (b) $\mathbf{Z}_1 \cup \mathbf{Z}_2 = \mathrm{Pa}(X) \cup \{Y\}$ where $Y\in\mathrm{Ch}(X)$, then when conditioned on $Y$, there exists an active (backdoor) path $\mathbf{B} \to \cdots \to \mathrm{Spouse}(X) \to Y \gets X$. Therefore, $X \perp (\mathbf{B} \setminus \mathbf{Z}_2) \mid (\mathbf{Z}_1 \cup \mathbf{Z}_2)$ does not hold.
> Hence, $\mathbb{V}[P_i(X\mid \mathbf{Z})] > 0$ because otherwise, $\mathbb{V}[P_i(X\mid \mathbf{Z})] = 0$ implies $X \perp (\mathbf{B} \setminus \mathbf{Z}_2) \mid (\mathbf{Z}_1 \cup \mathbf{Z}_2)$ which contradicts the above backdoor consequence.
>
> (c) $\mathrm{Pa}(X) \not\subseteq \mathbf{Z}_1 \cup \mathbf{Z}_2$ also implies the existence of an active backdoor connecting $\mathbf{B}$ and $X$. Using the same argument as in (b), we know this case also implies $\mathbb{V}[P_i(X\mid \mathbf{Z})] > 0$.
>
> Overall, (b) and (c) collectively ensure that as long as $\mathbf{Z}$ is not $\mathrm{Pa}(X)$, $\mathbb{V}[P_i(X\mid \mathbf{Z})] > 0$. Otherwise, (a) ensures that $\mathbb{V}[P_i(X\mid \mathbf{Z})] = 0.\quad \blacksquare$
>
> --
>
> We also note that the structural assumptions $\mathrm{Pa}(X) \cap \mathrm{Spouse}(X) = \emptyset$ and $\mathrm{Spouse}(X) \cap \mathbf{B} = \emptyset$ hold with relatively high probability. For example, regarding real-world graphs (provided by Bnlearn) we presented, the percentage of nodes satisfying these assumptions is over $75$ %.

---

> ### Author Response · Authors · 2025-11-07
> **Response to Reviewer Gmtf (3/3)**
>
> **Regarding the proof of Theorem 2**: We would like to explain that the proof of Theorem 2 shows that the size of any basis set is smaller than or equal to the size of the source set. But, the source set is also a valid basis set by definition (see Definition 3). This means the maximum size of a basis set is exactly the size of the source set. We have added this clarification to the end of the proof of Theorem 2. This has no critical effect on our logic flow in any way.
>
> ----
>
>
> **Regarding the proof of Lemma 1 of Theorem 3**:
>
> 3.1 Lemma 1 does not claim that “nodes with the fewest dependents” are always “sources”. Instead, it describes an either-or situation, summarized as follows:
>
> If a node $X$ has the smallest number of dependent nodes, then one of the following must hold:
>
> (a) $X$ is a source; or
>
> (b) $X$ depends on exactly one source.
>
> Let (p) = “$X$ has the smallest number of dependent nodes”. The original proof of Lemma 1 shows that “(p) and not (b)” leads to a contradiction. Hence, its logic negation, which is “(p) implies (b)”, must be true. As (b) implies (a), “(p) implies (b)” also means “(p) implies either (a) or (b)”.
>
> 3.2 Upon revision, we realize the argument about Type-I collider can be rewritten to induce a shorter proof as follows:
>     Suppose $X$ depends on two or more sources, i.e., (b) is false.
>     Then without conditioning on any observed evidence, there exist distinct sources $S_1$ and $S_2$ connected to $X$ via active (non-collider) trails, including causal, evidential, or common-cause.
>
> As a consequence, any node $Y$ dependent on either $S_1$ or $S_2$ (i.e., $Y \in \Phi(S_1) \cup \Phi(S_2)$) must also be dependent on $X$ (i.e., $Y \in \Phi(X)$).
> Hence, $\Phi(X) > \Phi(S1)$ and $\Phi(X) > \Phi(S2)$ because $S1$ cannot reach $S2$ without using a collider trail and hence, $S1 \notin \Phi(S2)$ and vice versa.
> Therefore, $X$ cannot have the smallest number of dependents, meaning "not (b)" implies "not (p)".
>     By contrapositive, we obtain "(p) implies (b)".
>
> ----
>
> **Regarding the choice for hyperparameters**: Table 3 shows that the two parameters of interest are the number of downsampled environments $m$ and the minimum downsampling ratio $\gamma$. Specifically, the higher value of $m$ shows noticeable overall improvements at the cost of proportionally increasing computational complexity. For practical implementation, we recommend choosing $m$ as high as the incurred increasing runtime is acceptable. As for $\gamma$, we find $\gamma = 0.5$ is an empirically reasonable selection. As a rule of thumb, $\gamma$ should not be too small so that estimations induced from downsampled data are adequately accurate, but it also needs to be small enough to promote diversity between downsampled data.

---

### Decision · Action_Editor_gmNr · 2025-12-15

**Recommendation:** Accept with minor revision

**Additional Comments:**

This paper studies a classical problem in causal inference: recovering the causal graph from observational data. This is a hard problem that cannot be done without some strong assumptions; moreover, many of the approaches have high sample or computational complexity. The authors introduce a new approach that is more computationally efficient than many alternatives, at least for a particular class of graphs. The basic idea is to use the variance of the effect/potential cause conditional distribution, using an invariance property. This is not hugely new, but the authors’ algorithmic application of the idea is interesting and appears to be effective, for at least quite a number of graphs.

The reviewers are largely favorable. This paper has some substantial improvements based on the reviewers’ careful read. It is now in stronger shape when it comes to specifying exactly what the assumptions are. Similarly the proofs are now much clearer. The reviewers (and I as well) find the technique interesting and creative. As a result, I think the paper should head for acceptance once some clarifications are added:

(i) The proof (discussed by the reviewers as well) of what is now Theorem 3 depends on Lemma 2; this proof should be expanded. Specifically, the step on the cardinality of $\Phi(S_1) \cup \Phi(S_2)$ uses the fact that $S_2 \not\in \Phi(S_1)$ to show that $|\Phi(S_1) \cup \Phi(S_2)| \geq |\Phi(S_1)|+1$. This step isn’t fleshed out enough: since $S_2 \not\in \Phi(S_2)$ (by definition of $\Phi$), we cannot immediately say whether there is any element in $\Phi(S_2)$ that is not in $\Phi(S_1)$ (which is what we need to give us the $ |\Phi(S_1)|+1$ lower bound). For this reason this step needs some reworking (or fleshing out if I am missing something).

(ii) The revised claim “this operation only removes information rather than generating new samples, it cannot introduce any spurious patterns into the data” should get a bit more justification.

There are also some minor typos (e.g., in 4.3, there is a “equation ??” left over) that should get corrected.

**Audience:**

Yes

**Audience Explanation:**

Yes, recovering causal graphs from data is a key area of causal inference and of wide inference to the TMLR audience.

**Claims And Evidence:**

Yes

**Claims Explanation:**

Yes, the authors introduce theoretical and empirical justifications for the claims made.

---

> ### Author Response · Authors · 2026-01-13
> **Follow up**
>
> Dear AE,
>
> We have updated the manuscript in which your comments are addressed.
>
> Specifically, for point 1, this is our typo. We want to clarify that X is always a member of $\Phi(X)$ by default. This can be verified in the code (line 106 in the file proposal.py of our code submission). The inclusion of $X\in\Phi(X)$ justifies that $|\Phi(S_1) \cup \Phi(S_2)| > |\Phi(S_1)| + 1$ since $S_2$ does not belong to $\Phi(S_1)$ and $S_2$ does belong $\Phi(S_2)$. This does not the correctness of our other theoretical results.
>
> For the second point, we will add the following elaboration after the claim "“this operation only removes information rather than generating new samples, it cannot introduce any spurious patterns into the data”: "Intuitively, downsampling can be viewed as introducing an auxiliary selection variable for each observational snapshot of the underlying DAG. Spurious dependence can arise only if this selection variable conditions on a collider of the form $U \to X \gets V$ where $U$ and $V$ originate from different source ancestries, in which case the corresponding trail changes from inactive to active. In our construction, however, downsampling uses $\boldsymbol{B} = \boldsymbol{b}$ as the sole selection criterion, so the selection variable depends only on basis variables. Each basis variable is either a source or a node with single-source ancestry (see Lemma 2). In the latter case, such a basis node can participate in a collider $U \to X \gets V$ only when $U$ and $V$ share the same source ancestry; otherwise, $X$ would depend on multiple sources and, by Lemma 2 and Theorem 3, could not be selected into the basis. Since $U$ and $V$ are already dependent via their common source, activating this collider does not introduce any new dependence."
>
> If you find the (still anonymous) updated manuscript meets the acceptance criterion, we will upload the camera-ready version with details of authors and acknowledgements.
>
> Best regards,
>
> Authors